# MULTI-OBJECTIVE TASK-AWARE PREDICTOR FOR IMAGE-TEXT ALIGNMENT

## ABSTRACT

Evaluating image-text alignment while reflecting human preferences across multiple aspects is a significant issue for the development of reliable vision-language applications. It becomes especially crucial in real-world scenarios where multiple valid descriptions exist depending on contexts or user needs. However, research progress is hindered by the lack of comprehensive benchmarks and existing evaluation predictors lacking at least one of these key properties: (1) *Alignment with human judgments*, (2) *Long-sequence processing*, (3) *Inference efficiency*, and (4) *Applicability to multi-objective scoring*. To address these challenges, we propose a plug-and-play architecture to build a robust predictor, MULTI-TAP (**Multi**-Objective **T**ask-**A**ware **P**redictor), capable of both multi and single-objective scoring. MULTI-TAP can produce a single overall score, utilizing a reward head built on top of a large vision-language model (LVLMs). We show that MULTI-TAP is robust in terms of application to different LVLM architectures, achieving significantly higher performance than existing metrics (*e.g.*, +42.3 Kendall's $\tau_c$ compared to IXCREW-S on FlickrExp) and even on par with the GPT-4o-based predictor, G-VEval, with a smaller size (7–8B). By training a lightweight ridge regression layer on the frozen hidden states of a pre-trained LVLM, MULTI-TAP can produce fine-grained scores for multiple human-interpretable objectives. MULTI-TAP performs better than VisionREWARD, a high-performing multi-objective reward model, in both performance and efficiency on multi-objective benchmarks and our newly released text-image-to-text dataset, EYE4ALL. Our new dataset, consisting of chosen/rejected human preferences (`EYE4ALLPref`) and human-annotated fine-grained scores across seven dimensions (`EYE4ALLMulti`), can serve as a foundation for developing more accessible AI systems by capturing the underlying preferences of users, including blind and low-vision (BLV) individuals. Our contributions can guide future research for developing human-aligned predictors.

## 1 INTRODUCTION

Accurate and efficient evaluation of image-text alignment is a fundamental task in multimodal research, serving as a key benchmark for assessing large vision-language models (LVLMs) (Lin et al., 2014; Hossain et al., 2019; Ghandi et al., 2023). As LVLMs are deployed in complex real-world scenarios, such as assistive technologies (Bandukda et al., 2019; Kazemi et al., 2023; Kuriakose et al., 2023; Chidiac et al., 2024) and instructional agents (Wang et al., 2024c; Li et al., 2024), the demand for human-aligned evaluation protocols for multimodal inputs has significantly increased (*e.g.*, automatic metrics (Grimal et al., 2024; Hartwig et al., 2024) and alignment training (Christiano et al., 2017; Schulman et al., 2017; Ahmadian et al., 2024)). Existing model-based image-text alignment predictors, also referred to as model-based metrics or reward models, can be categorized into three types: (a) encoder-based predictors, (b) text-based scoring predictors with generative LVLMs (generative reward models), and (c) scalar-based scoring predictors with generative LVLMs (scalar-based reward models). While each shows distinct strengths, none simultaneously satisfies four key properties : (1) *Strong correlation with human judgments*, (2) *Long-sequence processing*, (3) *Inference efficiency*, and (4) *Applicability to multi-objective scoring*. Meeting all four is essential for capturing diverse and context-dependent user preferences.

For example, (a) encoder-based predictors such as CLIP-Score (CLIP-S) (Hessel et al., 2021), BLIP-Score (BLIP-S) (Li et al., 2022), and related variants (Xu et al., 2024b; Sarto et al., 2023; Wada et al.,

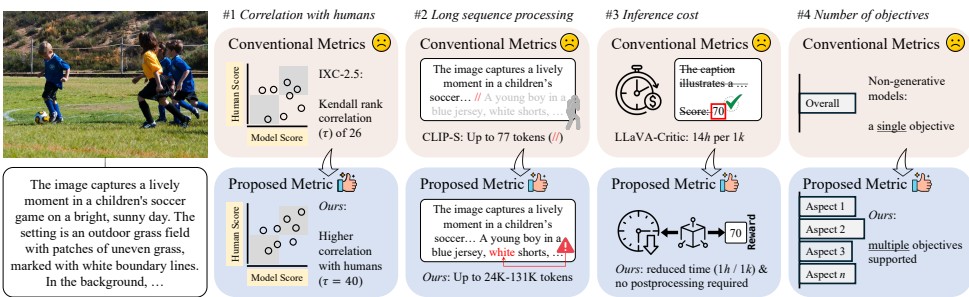

Figure 1: **Comparison between existing image-text alignment metrics and ours.** Our proposed method, applicable to different types of LVLMs, overcomes the challenges of conventional metrics in terms of (1) showing high correlations with human judgments, (2) understanding long input text sequences with detailed instructions, (3) reducing the inference time by returning precise scalar-based scores, and (4) enabling interpretable embeddings disentangled to multi-objective scores.

2024; An et al., 2024) are lightweight and efficient, yet their inherently limited context windows make understanding long text challenging (#2 in Figure 1). On the other hand, (b) generative reward models (Deitke et al., 2024; Xiong et al., 2024; Meta, 2024; Chen et al., 2024; Wang et al., 2024b) show improved semantic understanding supported by larger input size and alignment training. While effective, they are comparatively more computationally intensive (#3 in Figure 1) and often require prompt-tuning and additional post-processing (Li et al., 2024; Tong et al., 2024; Lambert et al., 2024), including bias subtraction to prevent ranking distortions (Zhu et al., 2025).

Finally, the most recent open-source *scalar*-based predictor, exemplified by InternLM-XComposer-2.5-Reward (IXCREW-S) (Zang et al., 2025), effectively addresses the aforementioned problems. Specifically, simply appending a scoring head on top of InternLM-XComposer-2.5 (InternLM) (Zhang et al., 2024c), it significantly improves cost efficiency and shows robust performance on the general vision-language reward benchmarks (Li et al., 2024). However, we observe weak agreement with the human judgments, captured by significantly low Kendall's $\tau$ on several image-text alignment benchmarks (Xu et al., 2019; Plummer et al., 2015) (#1 in Figure 1). Moreover, the model is released to be tied to a single LVLM backbone, InternLM, which constrains architectural modularity.

To address these limitations, we introduce a novel , LVLM-based predictor called MULTI-TAP (**Multi**-Objective **T**ask-**A**ware **P**redictor), capable of producing human-aligned scores for both single- and multiple-objective across diverse criteria (#4 in Figure 1). MULTI-TAP produces a robust single overall score and fine-grained scores aligned with multiple human-interpretable dimensions, utilizing the last hidden states from our newly trained reward model built on top of LVLM. Our predictor outperforms VisionReward (VisionREW-S) (Xu et al., 2024a), the only publicly available multi-objective scalar-based reward model for multimodal input (Xu et al., 2024a; Team, 2024). Here, to avoid confusion, "VisionREW-S/ImgREW-S" denotes models and "VisionREW/ImgREW" denotes datasets. Importantly, we instantiate the framework with widely used LVLMs, including Qwen2-VL (Wang et al., 2024b), InternLM (Zhang et al., 2024c), and LLaMA-3.2 (Meta, 2024), demonstrating model agnosticity, and at the same time, tackling all four core challenges in human-aligned evaluation.

To further validate our approach in a more challenging and practical setting, we introduce a novel text-image-to-text (TI2T) dataset, EYE4ALL, built upon judgments of 25 human annotators, including crucial perspectives from the blind and low-vision (BLV) individuals. Unlike existing datasets that focused on evaluating the quality of generated images (Xu et al., 2024a; Zhang et al., 2024e), EYE4ALL contains human judgments on the quality of the LVLM-generated *text response* and alignment to *text request* and *scenery image*. This unique BLV-centered benchmark is carefully curated, inspired by the BLV preference analysis from An et al. (2025). Specifically, the human annotators are guided to evaluate responses with respect to the BLV perspectives (given the BLV-driven request) rather than merely verifying the consistency of various image responses. For multiple evaluation purposes, we provide two complementary modes: `EYE4ALLPref`, consisting of the human preferences on two different LVLM text responses, and `EYE4ALLMulti`, a collection of human judgment scores across fine-grained dimensions, such as accuracy, sufficiency, and safety, facilitating both single- and multi-objective scoring evaluations. Our EYE4ALL covers diverse

pedestrian scenarios, enabling comprehensive assessment and systematic evaluation of recent LVLMs within realistic navigation contexts.

In summary, our study makes the following contributions: (1) **Scalable single- and multi-objective reward modeling framework on LVLM** for developing a robust scalar-based human-aligned predictor on multimodal datasets. (2) **MULTI-TAP**, strongly aligned with human judgments on both single and multiple dimensions. (3) **EYE4ALL**, a response-quality-oriented benchmark designed for practical evaluation and building robust assistive AI systems. Our work can serve an important role in guiding future research on robust and human-centered multimodal evaluation.

## 2 RELATED WORKS

### 2.1 IMAGE-TEXT ALIGNMENT EVALUATION PREDICTORS

Model-based metrics aim to automatically assess image-text alignment by approximating human judgment. The most widely used predictors are encoder-based metrics, which are commonly divided into *reference-based* and *reference-free*, where a reference denotes a human-written ground-truth caption paired with an image. Reference-based metrics (*e.g.*, Polos (Wada et al., 2024) and RefPAC-S (Sarto et al., 2024)) score the alignment between an image and a candidate caption conditioned on one or more references. In contrast, reference-free metrics (*e.g.*, CLIP-S (Hessel et al., 2021), BLIP-S (Li et al., 2022)) score the alignment based solely on image and candidate caption pair. Reference-based methods generally correlate more strongly with human ratings, but they require costly reference annotations. Inspired by the LLM-as-a-judge concept, recent work has explored generative LVLMs for evaluation (Deitke et al., 2024; Xiong et al., 2024; Meta, 2024). For instance, Tong et al. (2024) utilizes GPT-4o (OpenAI, 2024b) with the Chain-of-Thoughts (CoT) reasoning prompt to evaluate the alignment. In parallel, scalar-based reward models have also emerged by training a linear projection head on top of generative LVLMs (Zang et al., 2025). Nevertheless, existing multi-objective reward models, such as VisionREW-S (Xu et al., 2024a) and MPS (Zhang et al., 2024e), suffer from limited efficiency and accessibility.

### 2.2 IMAGE-TEXT ALIGNMENT EVALUATION DATASETS

Datasets for image-text alignment evaluation are typically categorized by two annotation settings: *pointwise* and *pairwise*. For the pointwise ranking datasets, each sample is labeled with the absolute human judgment scores across fine-grained scales (Xu et al., 2024b; Wada et al., 2024; Plummer et al., 2015), enabling nuanced metric analyses. Evaluations on these datasets often report Kendall's $\tau$ correlation to quantify alignment between metric scores and human judgments. Pairwise ranking datasets, in contrast, provide preference labels between two competing candidates (Xu et al., 2019; Shekhar et al., 2017). Instead of having absolute scores as the labels, the datasets are labeled as positive (human-preferred) or negative (human-rejected). Several pointwise ranking datasets, such as OID (Krasin et al., 2017) and Polaris (Wada et al., 2024), include multiple candidate captions with human judgment scores or ground-truth captions, enabling them to be repurposed for either pointwise or pairwise evaluations (An et al., 2025). Despite recent advances, few multimodal datasets are labeled with human judgment scores depending on varying preferences of users and criteria (Xu et al., 2024a; Team, 2024; An et al., 2025; Kang et al., 2025). This dataset scarcity hinders the development of LVLMs that can adapt their responses to varying contexts and user needs.

## 3 MULTI-OBJECTIVE TASK-AWARE PREDICTOR (MULTI-TAP)

We present MULTI-TAP (Figure 2), a robust image-text alignment predictor that supports single- and multiple-objective scoring. Our predictor can return to output either an overall score or multiple scores across their uniquely defined criteria. Training proceeds in two stages. At Stage 1, we train a single-objective predictor to produce a single, unified score that captures overall semantic alignment between images and texts, while simultaneously shaping semantically rich multimodal embeddings for Stage 2. During Stage 2, we use these frozen embeddings to build a *multi-objective, task-aware* predictor that produces scores across multiple human-interpretable dimensions. For inference, stage 1 is required for training to yield an overall image-text alignment score, while training both stages returns multiple scores across human-interpretable, fine-grained dimensions. To the best of our

knowledge, MULTI-TAP is the first human-aligned reward modeling framework explicitly designed for image-text alignment in accessibility-critical contexts.

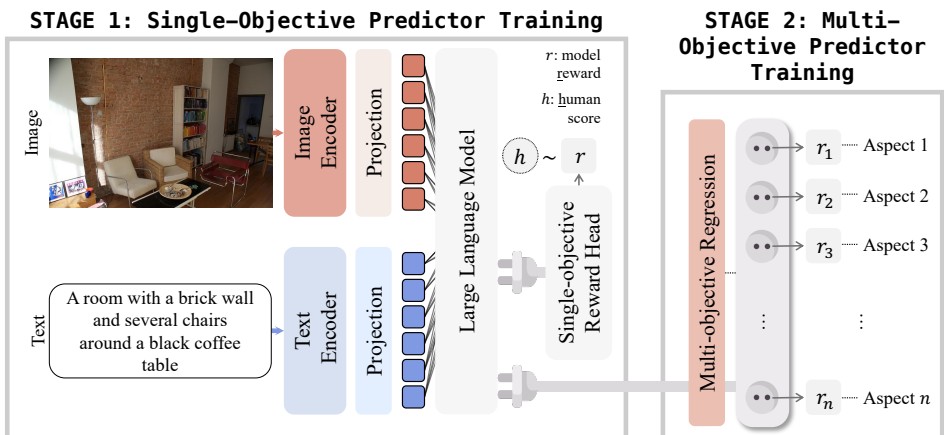

Figure 2: **Schematic diagram of proposed MULTI-TAP architecture**. At Stage 1, MULTI-TAP produces a scalar value reflecting image-text alignment by appending a reward head to the LVLM. For Stage 2, a ridge regression layer is added to the trained multimodal embeddings, generating scores across multiple aspects. We train the single-objective reward head and multi-objective regression layer in Stages 1 and 2, respectively.

### 3.1 SINGLE-OBJECTIVE PREDICTOR TRAINING

As shown in Figure 2, our architecture appends a reward head to the generative LVLM. This design allows the LVLM to process extended inputs and generate a unified semantically rich multimodal representation (#2 in Figure 1). The reward head maps the last hidden states of LVLM, or multimodal embeddings, to a *scalar* score, adaptable for single-objective scoring. It significantly reduces the inference latency compared to text-based generative scoring methods (#3 in Figure 1). We initialize the reward head using a zero-centered Gaussian distribution with standard deviation $\frac{1}{\sqrt{d+1}}$ ($d$: hidden dimension), following the standard initialization practices (von Werra et al., 2020). Departing from the Bradley-Terry (BT) model-based losses (Bradley & Terry, 1952) generally used in reward model training (Christiano et al., 2017; Stiennon et al., 2020), we adopt mean squared error (MSE) loss to explicitly align score outputs with human judgment scores: $\min_\theta \sum_i^N (r_i - h_i)^2$, where $\theta$ denotes parameters of LVLM and reward head, $r$ is the predicted scalar score, $N$ is the number of samples, and $h$ is the human judgment score. Our empirical findings indicate that MSE, with its convex formulation and simpler optimization, offers superior training stability and performance compared to the BT loss, where unstable spikes are frequently observed. We train LVLM and a reward model on two publicly available image-text alignment datasets: Polaris (Wada et al., 2024) and ImageReward (ImgREW) (Xu et al., 2024b), allowing the comprehensive alignment training of text quality judgment in terms of images and vice versa (details in Section 5). Training on these datasets helps the model to produce a robust score aligned with human judgment (#1 in Figure 1) and to create meaningful multimodal embeddings for multi-objective scoring (#4 in Figure 1).

### 3.2 MULTI-OBJECTIVE PREDICTOR TRAINING

Optimizing predictors directly under multi-objective settings is challenging. Prior work has shown that gradient-based optimization methods (*e.g.*, MGDA (Zhang et al., 2024d), PCGrad (Yu et al., 2020)) are costly at the scale of LVLMs and often fail to balance inherently conflicting objectives (He & Maghsudi, 2025). Ensemble-based approaches such as EMORL (Kong et al., 2025) partially alleviate these difficulties by aggregating models trained on individual objectives, but they still face instability and limited interpretability (without post-hoc calibration) when deployed at scale. These observations suggest that directly training a multi-objective predictor with LVLM is both expensive and unreliable.

To address this, we adopt a two-stage training paradigm. We reuse the multimodal embeddings from the previous stage by appending a multi-objective regression layer to the model, excluding

the original reward head. Inspired by previous work (Wang et al., 2024a), we treat the LVLM as a feature extractor (the plug symbol in Figure 2). Let $z_i \in \mathbb{R}^d$ denote the frozen multimodal embedding and let $y_i \in \mathbb{R}^K$ be the vector of human scores for $K$ dimensions. A ridge regression head predicts $\hat{y}_i = W z_i + b$ with parameters $W \in \mathbb{R}^{K \times d}$ and $b \in \mathbb{R}^K$ trained by $\min_{W,b} \sum_{i=1}^{N} \|y_i - W z_i - b\|_2^2 + \alpha \|W\|_F^2$. This head outputs multiple scalar scores per sample, each aligned with a human-interpretable criterion, improving transparency and supporting efficient customization (#4 in Figure 1). Utilizing precomputed hidden states from the frozen LVLM backbone, these regression heads can be trained asynchronously to meet task-specific needs. We opt for ridge regression as a principled design choice to maximize model interpretability, efficiency, and scalability, especially in deployment scenarios where full finetuning or gradient-based optimization is prohibitive.

Unlike ArmoRM and VisionREW-S (Xu et al., 2024a), which aggregate multi-objective outputs into a single score through a learned linear or gating head, we do not aggregate multi-objective outputs into a single overall score. This is because score aggregation across predefined dimensions may fail to holistically capture overall quality. For instance, VisionREW-S has several zero values in its aggregation head, which means the weights for specific dimensions are unstable and uninformative for computing an overall score. Instead, we use a single overall reward score from Stage 1, and a set of dimension-specific scores derived via lightweight ridge regression training from Stage 2.

## 4 EYE4ALL: BLV-AWARE, MULTI-OBJECTIVE IMAGE-TEXT ALIGNMENT SCORING DATASET

To supplement the limited pool of current multi-objective scoring datasets containing fine-grained human judgment scores for image-text alignment evaluation and to validate our MULTI-TAP in a more realistic setting, we introduce EYE4ALL, a curated evaluation dataset. This section outlines the LVLM generation pipeline and the human annotation protocol (detailed steps are in Appendix A).

### 4.1 LVLM RESPONSE COLLECTION

We first collect diverse text responses from LVLMs conditioned on an image and a scene-relevant request. Existing multi-objective image-text alignment scoring datasets (Xu et al., 2024a; Team, 2024; Zhang et al., 2024e) typically rely on judgment scores concerning image quality and emphasize generic preferences. In contrast, EYE4ALL consists of reasoning chains that are useful and applicable to (but not limited to) Blind and Low-Vision (BLV) users for navigational purposes in daily lives. We use Sideguide (Park et al., 2020) and Sidewalk (AIHub, 2019) scenery image corpora, pairing each image with BLV-plausible textual requests aligned to the depicted scene (An et al., 2025).

We collect text responses by prompting QWEN2-VL (Wang et al., 2024b), LLaVA-1.6 (Liu et al., 2023), and InternLM-XComposer2-VL (InternLM-X2-VL) (Dong et al., 2024), which are known to demonstrate remarkable in-context learning ability (Zong et al., 2025). These models are instructed to generate responses from the perspective of BLV users, utilizing the model instructions and the BLV-plausible requests from An et al. (2025). For each instance, we randomly select one of the three different LVLM responses and refine the responses by prompting GPT-4o mini (OpenAI, 2024b) the original LVLM responses (system and few-shot prompts are in Appendix B). This process of using GPT-4o to refine responses from various models rather than generating them from scratch helps to minimize the introduction of strong bias induced by a single model.

### 4.2 HUMAN ASSESSMENT FOR EVALUATING LVLM RESPONSES

We collect human judgments from 25 sighted human annotators (approved by the Institutional Review Board), evaluating the collected LVLM responses in terms of two main aspects: (1) whether the text response aligns with the image and the text request, and (2) whether the response addresses potential safety concerns evident in the scenery image that BLV users might otherwise miss. To balance efficiency and diversity of annotations, we randomly sample 1k image-request-response triplets from the previous stage. Each annotator completes 100 items in an online setting, typically taking 2 to 3 hours.

For each sample, the annotators are required to rate the refined GPT-4o responses along seven dimensions: (1) *Direction Accuracy*, (2) *Depth Accuracy*, (3) *Safety*, (4) *Sufficiency*, (5) *Conciseness*, (6) *Hallucination*, and (7) *Overall Quality*. These criteria were selected based on the importance and challenges of producing safe and informative LVLM responses given the image and request (Karamolegkou et al., 2025). The accuracy criterion is divided into two aspects (*Direction* and *Depth*) since precise spatial guidance directly affects user safety, particularly for BLV users. In addition, we exclude requests that contain horizontal directions beyond the 9 to 3 o'clock, since directions from 4 to 8 o'clock correspond to areas behind the viewer and are not visible in standard non-panoramic images. Unlike the other criteria, the *Hallucination* is assessed in a dichotomous format to capture misleading or false navigational content. This is distinct from *Safety*, where the focus is on whether the response includes obstacles or provides safety-relevant guidance.

All aspects are annotated with an averaged scalar score from 2–3 human judgment scores on a Likert scale of 1 to 5, except for the *Hallucination*, which is evaluated as either 0 (presence) or 1 (absence), depending on these three issues: (1) non-related information, (2) inaccurate step-by-step order, and (3) repeated content. The other six dimensions are evaluated with a fine-grained rubric (details in Appendix B). The human annotation procedure yields 2,112 unique samples that constitute the `EYE4ALLMulti` dataset for multi-objective scoring assessment. In addition, we also collect high-quality human-refined captions for each image (and request)-response pair that satisfy all seven evaluation criteria, labeled as positive (preferred) samples in our human preference dataset, `EYE4ALLPref`. The well-constructed predictors should understand the scenery images as well as context-dependent requests of our EYE4ALL to simulate human judgment preferences and fine-grained scores across multiple dimensions (more details are in Appendix B).

## 5 EXPERIMENTS

We evaluate MULTI-TAP through comprehensive experiments spanning both single- and multi-objective settings. Our study benchmarks against a wide range of predictors and datasets, enabling a rigorous assessment of its robustness and efficiency across diverse tasks and model architectures.

### 5.1 MULTI-TAP TRAINING

**Datasets.** We train the single-objective predictor using two open-sourced datasets with complementary annotation protocol: Polaris (Wada et al., 2024) and ImageReward (ImgREW) (Xu et al., 2024b). Polaris is a dataset containing annotations of captions based on given images; on the other hand, ImgREW is a dataset evaluating the quality of generated images according to prompts. The Polaris dataset consists of general image-caption alignment scores ranging from 0 to 1 (discretized into 0.0, 0.25, 0.5, 0.75, 1.0). ImgREW also consists of scores, ranging from 1–7 (later normalized to 0–1), which measure prompt-image alignment. Note that these two datasets are all open-source, and we adhere to the predefined training/test splits, following standard practice.

**Models.** To examine the generalization across model scales and architectures, we instantiate MULTI-TAP on the following widely used LVLMs: Qwen2-VL-2B/7B (Wang et al., 2024b), InternLM-XComposer-2.5-7B (InternLM-7B) (Zhang et al., 2024c), and LLaMA-3.2-11B (Meta, 2024). For the 2B model, we set the learning rate to 2e-7, and for the larger models, we use a learning rate of 2e-6. We utilize 8 A100 GPUs for training LLaMA-3.2-11B and 8 RTX A6000 GPUs for the others with seed 42. All models are trained for a single epoch, with a batch size of 8 and a gradient accumulation of 4. For the multi-objective version of MULTI-TAP, the backbone LVLM is frozen, and only a lightweight ridge regression head over the final hidden states is trained. We perform a hyperparameter search of the regularization coefficient $\alpha$ within the scope of $\{0.001, 0.01, 0.1, 1, 10, 100\}$, selected based on the lowest training loss. We train MULTI-TAP with a single epoch for each $\alpha$ since we empirically observe rapid and stable convergence during the first epoch (for both stages).

### 5.2 MULTI-TAP EVALUATION

**Datasets.** We evaluate the efficacy of MULTI-TAP across diverse image-text alignment benchmarks: PASCAL-50S (Xu et al., 2019), FOILR1/R4 (R1 and R4 refer to evaluation using one and

| | **Pairwise Ranking Datasets** | | | | | | **Pointwise Ranking Datasets** | | |
|---|---|---|---|---|---|---|---|---|---|
| | **PASCAL** | **FOILR1** | **FOILR4** | **Polaris*** | **OID*** | **ImgREW** | **FlickrExp** | **FlickrCF** | **Polaris** |
| | P-Acc | P-Acc | P-Acc | P-Acc | P-Acc | P-Acc | $\tau_c$ | $\tau_b$ | $\tau_c$ |
| *CLIP/BLIP-based predictors* | | | | | | | | | |
| CLIP-S | 80.7 | 87.2 | 87.2 | 79.7 | 56.5 | 56.7 | 51.2 | 34.4 | 52.3 |
| LongCLIP-S | 82.8 | 91.6 | 91.6 | 77.5 | 58.1 | 56.5 | 54.1 | 35.4 | 54.0 |
| PAC-S | 82.4 | 93.7 | 94.9 | 77.0 | 57.7 | 57.2 | 55.9 | 37.6 | 52.5 |
| Ref-free Polos | 81.0 | 88.7 | 88.7 | 60.0 | 66.2 | 56.6 | 51.4 | 34.4 | 52.3 |
| RefCLIP-S† | 83.1 | 91.0 | 92.6 | - | - | - | 53.0 | 36.4 | 52.3 |
| RefPAC-S† | 84.7 | 88.7 | 94.9 | - | - | - | 55.9 | 37.6 | 56.0 |
| Polos† | **86.5** | 93.3 | 95.4 | - | - | - | 56.4 | 37.8 | 57.8 |
| BLIP-S | 82.5 | 95.1 | 95.1 | 79.5 | 59.3 | 57.8 | 57.1 | 37.8 | 54.0 |
| ImgREW-S | 81.5 | 93.8 | 93.8 | 73.3 | 58.5 | **65.2** | 49.8 | 36.2 | 52.3 |
| *Reward model-based predictors* | | | | | | | | | |
| IXCREW-S | 74.2 | 94.3 | 94.3 | 81.9 | 57.5 | 53.6 | 17.0 | 25.7 | 50.5 |
| **MULTI-TAP** | | | | | | | | | |
| - Qwen-2B-S | 81.5 | **98.0** | **98.0** | **87.0** | 53.2 | 59.0 | 56.8 | 38.9 | 60.1 |
| - Qwen-7B-S | 84.0 | 97.8 | 97.8 | 82.2 | 58.1 | 63.2 | 58.1 | 38.5 | 61.1 |
| - InternLM-7B-S | 83.2 | 96.5 | 96.5 | 81.6 | 61.0 | 61.6 | **59.3** | **39.5** | **61.6** |
| - LLaMA-3.2-S | 83.0 | 96.9 | 96.9 | 78.8 | **68.7** | 62.2 | 56.8 | 38.0 | 60.7 |

Table 1: **Performances of various predictors (S: Score) on image-text alignment datasets**. For both pairwise and pointwise ranking evaluation, MULTI-TAP models consistently outperform other metrics. Reference-based metrics, marked with †, cannot be evaluated on datasets without references (indicated by "-").

four references) (Shekhar et al., 2017), Polaris* (Wada et al., 2024), OID* (Krasin et al., 2017), ImageReward (ImgREW) (Xu et al., 2024b), Flickr8k-Expert (FlickrExp) and Flickr8k-CF (FlickrCF) (Plummer et al., 2015), and Polaris (Wada et al., 2024). An asterisk (*) indicates that the original dataset has been reformulated into a pairwise comparison format by binarizing scores at the median threshold to separate preferred from rejected samples. For OID*, we use a curated 246-sample subset of exact matches due to partial availability. The first five pairwise ranking datasets are evaluated with pairwise accuracy (P-Acc), where a higher score for the positive sample indicates a correct answer. The pointwise ranking datasets are evaluated with Kendall's correlation coefficients ($\tau_b$ or $\tau_c$), scaled by 100 for comparability with accuracy. To further evaluate performance on long-form and diverse prompts (image-to-text; I2T, text-to-image; T2I, and text-image-to-text; TI2T), we also test on Sightation (Kang et al., 2025), Align-anything (Team, 2024), and our EYE4ALLPref.

For multi-objective scoring evaluation, we benchmark predictors on VisionREW (Xu et al., 2024a), Align-anything (Team, 2024), and our EYE4ALLMulti datasets. Since the test set of VisionREW is not publicly available, we randomly sample 1k samples from its training data for evaluation and use the remainder for training. For EYE4ALLMulti, we construct an additional 1k training set comprising scene-request pairs and responses of GPT-4o mini (OpenAI, 2024b). This dataset results in a diverse score range for each criterion, avoiding model overfitting for one particular dimension. Since VisionREW-S only outputs binary-scaled scores, we apply median-based binarization to Align-anything (TI2T-Binary, T2I-Binary) and EYE4ALLMulti (*e.g.*, using a threshold of 2 on a 1–4 scale). MULTI-TAP, in contrast, outputs continuous scores; we report the uncalibrated multi-objective scores as the baselines (*see* Figure 10 in Appendix C).

**Comparison Baselines.** In the single-objective setting, we compare MULTI-TAP with a broad spectrum of predictors. First, we include the CLIP and BLIP-based metrics: CLIP-S/RefCLIP-S (Hessel et al., 2021), LongCLIP-S (Zhang et al., 2024a), PAC-S/RefPAC-S (Sarto et al., 2023), Ref-free Polos/Polos (Wada et al., 2024), BLIP-S (Li et al., 2022), and ImageReward (ImgREW-S) (Xu et al., 2024b). We also report reference-based metrics run in a reference-free configuration. In addition, we include generative reward models and scalar-based reward models: Molmo-7B (Deitke et al., 2024), LLaVA-Critic-7B (Xiong et al., 2024), Qwen2-VL-7B (Wang et al., 2024b), InternVL2-8B (Chen et al., 2024), and IXCREW-S (Zang et al., 2025). Lastly, we compare ours with G-VEval (Tong et al., 2024), which leverages GPT-4o mini (OpenAI, 2024b). To compare MULTI-TAP with existing multi-objective models, we select VisionREW-S (Zhang et al., 2024b), the only publicly available multimodal multi-objective scalar-based predictor to our knowledge. We employ the BF16 release with default settings and disable the masking method for better performance in our runs.

# 6 RESULTS

## 6.1 CORRELATION WITH HUMAN JUDGMENTS

We first show that our proposed MULTI-TAP models generally align well with human judgments in terms of *both* pairwise and pointwise ranking datasets. As shown in Table 1, four versions of MULTI-TAP, built on different LVLMs, mainly outperform conventional CLIP-, BLIP-, and reward model-based predictors across a wide range of image-text alignment benchmarks. Regardless of the architectures, MULTI-TAP generally achieves higher performances than the existing scalar-based reward model, IXCREW-S. In particular, MULTI-TAP$_{Qwen-2B-S}$ notably shows the best accuracy performances on FOIL and Polaris*, achieving 98.0% and 87.0%. On top of that, our predictor aligns significantly better in terms of human judgment rank correlations ($\tau$s), where MULTI-TAP$_{InternLM-7B-S}$ achieves the highest performances (*e.g.*, 59.3, 39.5, and 61.6 on FlickrExp, FlickrCF, and Polaris). Hence, MULTI-TAP generally attains the best performance across diverse image-text alignment datasets, exhibiting high correlations on pointwise ranking datasets (compared to CLIP/BLIP-based predictors) and pairwise ranking datasets (compared to the SoTA reward model-based predictor). The superior performance of MULTI-TAP compared to other predictors underscores the robustness of our predictors in capturing correlations with human judgments.

## 6.2 LONG-SEQUENCE PROCESSING

Table 2 demonstrates that MULTI-TAP models are also superior in understanding long and diverse formats of input prompts, attributable to the inherent long sequence understanding capability of LVLMs. Our predictors perform strongly on the conventional image-to-text (I2T) task, as well as on text-to-image (T2I) and text-image-to-text (TI2T) settings. MULTI-TAP$_{InternLM-7B-S}$ and MULTI-TAP$_{Qwen-7B-S}$ achieve the highest accuracies on Sightation (I2T) and Align-anything (T2I), re-

| | *Max Token #* | Sightation | Align-anything | | EYE4ALLPref |
|---|---|---|---|---|---|
| | | I2T | T2I | TI2T | TI2T |
| CLIP-S | 77 | 42.8 | 63.7 | 50.7 | 34.6 |
| LongCLIP-S | 248 | 49.1 | 64.7 | 48.7 | 17.9 |
| BLIP-S | 512 | 48.3 | 49.2 | 53.1 | 44.6 |
| IXCREW-S | 24k | 51.4 | 54.7 | **74.4** | **78.3** |
| **MULTI-TAP** | | | | | |
| - Qwen-2B-S | 32k | 50.2 | 59.8 | 47.4 | 40.7 |
| - Qwen-7B-S | 32k | 49.4 | **71.5** | 60.5 | 64.1 |
| - InternLM-7B-S | 24k | **53.1** | 64.6 | 57.0 | 57.2 |
| - LLaMA-3.2-S | 131k | 47.9 | 71.3 | 54.3 | 59.4 |

Table 2: **Maximum tokens per input and performances of metrics on multimodal data with long contexts**. MULTI-TAP shows strong capability in human preference alignment, especially for I2T and T2I tasks.

spectively. Although our predictors do not surpass IXCREW-S in the two TI2T datasets, possibly due to the lack of large-scale training data in TI2T format (our training data is in I2T/T2I format, and training data of IXCREW-S are not publicly disclosed), they show significantly improved performances on pointwise ranking datasets (Table 1). Moreover, the consistent ordering of systems on Align-anything and our `EYE4ALLPref` supports both the validity of our modeling framework and the practical relevance of the proposed dataset.

## 6.3 INFERENCE EFFICIENCY

We also compare the performances of the generative reward models with ours in Table 3. While these generative models excel in standard tasks, there remain limitations to applying them as predictors in two respects: (1) extensive inference time (*e.g.*, at least 90 hours for InternVL2-8B on Polaris* dataset) and (2) significantly low human correlation performances on pointwise ranking datasets. Due to the extensive inference time of generative reward models, we evaluate predictors on 100 samples except for Polaris*

| | *Time (hrs)* | FlickrExp | | FlickrCF | | Polaris* | ImgREW |
|---|---|---|---|---|---|---|---|
| | | P-Acc | $\tau_c$ | P-Acc | $\tau_b$ | P-Acc | P-Acc |
| Molmo-7B | 50 | 40.0 | 2.35 | 49.0 | 20.6 | 54.0 | 20.0 |
| Qwen2-VL-7B | 42 | 70.0 | NaN | 69.0 | NaN | 49.9 | 1.02 |
| LLaVA-Critic-7B | 28 | 80.0 | 10.7 | 91.0 | 26.7 | 76.0 | 37.5 |
| InternVL2-8B | 90 | 95.0 | 22.6 | 91.0 | 10.7 | 77.4 | 50.4 |
| LLaMA-3.2-11B | 6 | **100.0** | 5.29 | **100.0** | 9.00 | 85.9 | 51.6 |
| **MULTI-TAP** | | | | | | | |
| - Qwen-2B-S | 1.5 | 94.0 | 37.6 | 86.0 | 20.8 | 81.7 | 60.6 |
| - Qwen-7B-S | 2 | **100.0** | **54.7** | 99.0 | **30.3** | 82.2 | **63.2** |
| - InternLM-7B-S | 2 | **100.0** | 46.6 | 99.0 | 29.4 | **87.0** | 59.0 |
| - LLaMA-3.2-S | 5.5 | **100.0** | 52.1 | 99.0 | 28.4 | 78.8 | 62.0 |

Table 3: **Performances of generative reward models and ours on image-text alignment datasets**. MULTI-TAP shows robust performances with significantly reduced inference time (measured for Polaris dataset).

and ImgREW, where we use the entire test set ($n$ = 14k and 466). Although InternVL2-8B shows high accuracies on Flickr, the correlations are significantly low (*e.g.*, $\tau_c$ =22.6 on FlickrExp), compared to those of MULTI-TAP.

In contrast, MULTI-TAP performs well on both pairwise and pointwise benchmarks. Notably, MULTI-TAP$_{Qwen-7B-S}$ achieves the highest correlation coefficients on both FlickrExp and FlickrCF ($\tau_c$ = 54.7 and $\tau_b$ = 30.3), while maintaining near-perfect preference accuracies.

Additionally, our predictors show superior inference efficiency compared to generative reward models (*e.g.*, up to 1.5 hours for MULTI-TAP$_{\text{Qwen-2B-S}}$ on the Polaris* dataset). As shown in Table 4, our predictors also achieve on-par performance with G-VEval, without API cost overhead, attaining the highest $\tau_c$ of 59.3 on FlickrExp and the same best accuracy of 97.8 on FOILR1, highlighting their effectiveness as alignment predictors. We omit the scores of G-VEval on FOIL due to their unavailability and extensive cost.

| | Open? | # Param | Flickr Expert $\tau_b$ | Flickr Expert $\tau_c$ | FOILR1 P-Acc | FOILR4 P-Acc |
|---|---|---|---|---|---|---|
| G-VEval | ✗ | ~ 200B | | | | |
| - w/o CoT prompt | ✗ | ~ 200B | 50.2 | 48.4 | - | - |
| - w/o reason | ✗ | ~ 200B | 52.4 | 26.9 | - | - |
| - w/o expected score | ✗ | ~ 200B | 59.1 | 54.9 | - | - |
| - full setting | ✗ | ~ 200B | **60.4** | 58.6 | 97.8 | **98.4** |
| **MULTI-TAP** | ✓ | | | | | |
| - Qwen-2B-S | ✓ | 2B | 55.2 | 56.8 | 93.2 | 93.2 |
| - Qwen-7B-S | ✓ | 7B | 57.7 | 58.1 | **97.8** | 97.8 |
| - InternLM-7B-S | ✓ | 7B | 58.9 | **59.3** | 96.5 | 96.5 |
| - LLaMA-3.2-S | ✓ | 11B | 56.5 | 56.8 | 96.9 | 96.9 |

Table 4: **Comparison between G-VEval and ours on image-text alignment datasets**. Our open-sourced, MULTI-TAP achieves performances comparable to the GPT-4o-based predictor with fewer model parameters.

## 6.4 MULTI-OBJECTIVE SCORING

We demonstrate the effectiveness of MULTI-TAP on multi-objective datasets, including our proposed `EYE4ALLMulti` with comparison to a SoTA multi-objective reward model, VisionREW-S.

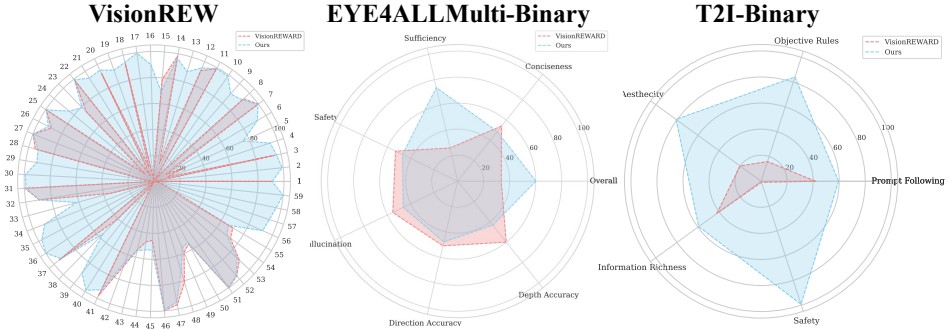

Figure 3: **Performances of VisionREW-S (red) and our MULTI-TAP (blue) on multi-objective datasets**. Our MULTI-TAP$_{\text{Qwen-7B-S}}$ generally outperforms VisionREW-S (19B), achieving 34%p, 3%p, 53%p higher accuracies on VisionREW, `EYE4ALLMulti`-Binary, and Align-anything (T2I-Binary) datasets.

As illustrated in Figure 3, on the VisionREW held-out training set ($n = 1k$), our MULTI-TAP consistently achieves an average accuracy of at least 87.2 across all dimensions, significantly outperforming the average score of 53.3 from VisionREW-S. Moreover, our predictor shows high performance across 59 dimensions, whereas VisionREW-S tends to overfit on specific dimensions since it relies on an aggregation head, where its performance depends on learned weights. Additionally, VisionREW-S requires separate inference for each dimension, resulting in a total inference time of 51 days on a single RTX A6000. In contrast, our predictor completes both training and inference in about 4 hours for MULTI-TAP$_{\text{Qwen-2B-S}}$ and in about 11 hours for MULTI-TAP$_{\text{LLaMA-3.2-S}}$.

On Align-anything datasets, MULTI-TAP achieves the least average binary classification accuracy of 94.07 for TI2T and 75.58 for T2I tasks, whereas VisionREW-S achieves only 5.47 and 24.05, respectively. Evaluated on a finer scale in the range of 1–4, where VisionREW-S cannot be operated due to output setting, MULTI-TAP shows robust performance, achieving at least 53.88 and 50.96 on TI2T and T2I tasks. Finally, on our `EYE4ALLMulti` benchmark, MULTI-TAP not only achieves the best 52.08, surpassing VisionREW-S performance of 47.63, but also

| Model Type | Accuracy (%) | Inference Complexity |
|---|---|---|
| MLP | **88.6** | $O(\sum_{l=1}^{L} n_{l-1} n_l)$ |
| Random Forest | 88.2 | $O(T \cdot D)$ |
| Ridge Regression | 87.2 | $O(d)$ |

Table 5: **Comparison of VisionREW performance (59 dimensions) across model types for the second stage.** Ridge regression provides the lowest inference complexity while maintaining competitive accuracy compared to MLP and Random Forest (Abbreviations: $d$: input feature dimension, $L$: number of layers, $n_l$: number of hidden units in layer $l$, $T$: number of trees, $D$: maximum tree depth.).

shows robust performance with at least 36.36 on the fine-grained scale (details in Appendix C). Finally, the ablation results of replacing ridge regression with MLP or Random Forest, as shown in Table 5, suggest that using ridge regression significantly reduces the inference cost while pre-

serving accuracy performance. These results underscore the reliability of `EYE4ALLMulti` and MULTI-TAP as a rigorous benchmark and predictor, respectively.

## 7 CONCLUSION

As multimodal models rapidly evolve, there is a growing need for automatic evaluation metrics beyond traditional rule-based approaches that capture coarse semantic similarity, yet they struggle with long, instruction-rich texts that reflect real-world scenarios. While generative reward models offer improved semantic alignment and long-text understanding, they are limited in practicality due to high computational costs. To address these challenges, we introduce MULTI-TAP, a multi-objective-supported predictor built upon generative LVLMs. The proposed stage 1 training yields meaningful multimodal embeddings that are utilized in the later stage to build a predictor capable of multi-objective scoring with better robustness and efficiency. In addition, we introduce an extensively curated human-validated dataset EYE4ALL, designed to benchmark evaluation metrics through pairwise preferences (`EYE4ALLPref`) and fine-grained, pointwise preference scores across multiple dimensions (`EYE4ALLMulti`). Our released dataset will significantly contribute to the limited multimodal data pool annotated with multi-objective scores, spanning diverse human-interpretable criteria. Future studies could advance LVLMs that incorporate the needs of people with accessibility needs using our robust multi-objective task-aware predictor.

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

APPENDIX

Due to the limited pages, we provide supplementary materials in the Appendix with the following contents:

- Section A: Model Prompt Details
- Section B: Dataset Benchmark Details
- Section C: Additional Results
- Section D: Limitations and Broader Impacts

## A  MODEL PROMPT DETAILS

We provide all the prompts for the evaluation using generative large vision-language models (LVLMs) since the generative models require detailed instruction prompts for the text-based score generation. Table 6 shows the prompts for the evaluation using Molmo-7B Deitke et al. (2024), Qwen2-VL-7B Wang et al. (2024b), and InternVL2-8B Chen et al. (2024) for both pairwise (above) and pointwise (below) ranking tasks. Tables 7 and 8 show the prompts for the evaluation using LLaVA-Critic-7B Xiong et al. (2024) and LLaMA-3.2-11B Meta (2024). Lastly, we use two variations for prompting IXCREW-S Zang et al. (2025) (Table 9), where the results are in Appendix C.

## B  DATASET BENCHMARK DETAILS

**LVLM generation**  After the response generation using in-context learnable 7B models mentioned in Section 4 of the paper, we refine the LVLM responses using GPT-4o mini ($0.15/1M input tokens) OpenAI (2024b) using the prompt in Table 10. Furthermore, Table 11 shows the prompt used to generate scores across seven dimensions in constructing the EYE4ALL training dataset.

**Human study**  The human experiment was approved by the Institutional Review Board (IRB). The annotation guideline distributed to all annotators can be seen in Table 12. The sample screenshot of each question is in Figure 4. We encouraged the annotators to actively ask any questions on technical or ambiguous/confusing problems. If there were overlapping questions, we notified all the annotators to make sure the annotations were consistent. Examples of EYE4ALLMulti and EYE4ALLPref are illustrated in Figures 5 and 6, respectively.

The distribution of leading time and length of newly added captions is in Figure 7. The first row of Figure 7 visualizes the plots per *question*, and the plots in the second row are for each *annotator*. According to the correlation plot in the third column of the second row, more time spent in the annotation does not necessarily mean more lengthy captions added per annotator. The average and standard deviation of the annotator agreement are 33.21 and 17.70.

**EYE4ALL distribution**  The summarized results of human-annotated scores evaluated from seven different perspectives are presented in Figure 8. We observe that current LVLMs Xiong et al. (2024); Wang et al. (2024b); Dong et al. (2024), including GPT-4o mini OpenAI (2024b), are not entirely reliable in generating responses with precise direction and depth information. Although most responses are regarded as "entirely safe and actionable," the LVLM responses that include more than one inaccurate direction/depth information are critical for the BLV users in navigation. Thus, the accuracy aspect would be one main challenge for LVLMs to be directly applicable in assistant technologies.

A common observation in the aspect of *sufficiency* is that human annotators mostly disagreed with the notion that the LVLM responses were sufficient, different from a higher agreement for the *conciseness* category. While sufficiency is inherently subjective, depending on whether annotators believe the response provides all necessary information for BLV users to complete a task (as defined in our guidelines), this particular dimension shows the highest correlation with overall ratings (0.85 in Figure 9), indicating that the LVLM responses should be sufficient to reach high overall ratings from humans. In contrast, the *hallucination* category exhibits the lowest correlation (0.35), which may stem from differences in scoring scales. Nevertheless, nearly 1k responses identified

instances of hallucination, indicating that this issue remains prevalent and requires close monitoring. Consequently, these findings highlight the need for LVLMs to further improve their ability to generate a comprehensive, task-relevant context for BLV users.

## C  ADDITIONAL RESULTS

**MULTI-TAP performances using different prompt format**    Table 13 shows the performances of MULTI-TAP when trained with the prompt format of LLaMA-3.2 Meta (2024), which omits explicit instructions (*i.e.*, uses an empty prompt). This experiment evaluates the flexibility and robustness of the reward model in scenarios where no task-specific guidance, even if the instruction (*i.e.*, 'Describe the image.') was not explicitly given to the model. Compared to the results of Table 1 in the paper, we observe subtle performance differences, yet the overall evaluation trends remain consistent. For instance, higher correlation coefficients and lower preference accuracy (84.7 in Table 13 *vs.* 81.5 in Table 1 for PASCAL-50S) performances are achieved using the empty prompt setting in MULTI-TAP$_{\text{Qwen-2B-S}}$.

**MULTI-TAP performances on multi-objective scoring datasets**    Figure 10 presents the performance of MULTI-TAP on multi-objective scoring datasets with fine-grained scales: a 1–4 scale for Align-anything (TI2T and T2I) Park et al. (2020) and a normalized 1–5 scale for `EYE4ALLMulti`, where each raw score in the range of 1–5 was averaged across 2–3 annotators. Due to the constraint of the VisionREW-S Xu et al. (2024a) that can only output binary scores (by answering yes or no), we can only provide performances of our MULTI-TAP models on the original Align-anything and `EYE4ALLMulti` datasets. The performances of VisionREW-S and MULTI-TAP models for every dimension can be examined in Tables 14, 15, 16, and 17 on VisionREW Xu et al. (2024a), `EYE4ALLMulti`-Binary, and Align-anything T2I-Binary datasets. We consistently surpass the dimension-level performance of VisionREW-S, especially in VisionREW and Align-anything datasets.

**Generative model performances**    Tables 18, 19, and 20 show sample responses of generative LVLMs: Molmo-7B Deitke et al. (2024), LLaVA-Critic Xiong et al. (2024), and InternVL2-8B Chen et al. (2024). Since the answers include reasons for their choice or scores, we extract the final scalar ratings via post-processing: specifically, the floating-point value following "Overall Judgment" for Molmo-7B and InternVL2-8B, and the values after "The better caption" and "Score" for LLaVA-Critic-7B. Additionally, Tables 21 and 22 report the performances under alternative prompting strategies for IXCREW-S Zang et al. (2025) and LLaMA-3.2-11B Meta (2024), respectively.

## D  LIMITATIONS AND BROADER IMPACTS

While MULTI-TAP demonstrates strong alignment with human scores in both pairwise and pointwise ranking tasks with notable efficiency and the performance is consistent across different training seeds, its performance on very challenging multi-objective benchmarks such as EYE4ALL remains limited. Although MULTI-TAP outperforms existing open-source multi-objective state-of-the-art models, achieving consistently high accuracies across diverse evaluation dimensions is still an open challenge. We leave as future work the development of LVLMs that can generate more accurate, hallucination-free, and human-preferred responses, particularly for demanding multi-objective evaluation settings. From a broader perspective, MULTI-TAP contributes to building human-centered multimodal evaluation frameworks. By releasing EYE4ALL, we aim to support the creation of robust assistive AI systems—especially for Blind and Low-Vision users—while underscoring the need for transparent, interpretable, and efficient evaluation. Nonetheless, careful deployment is necessary to mitigate risks such as hallucination, bias, or unsafe outputs, ensuring that these technologies maximize societal benefit while minimizing unintended harms. We acknowledge the use of the LLM, GPT-5 and Gemini 2.5-Flash in writing the paper, for grammatical and contextual correction purposes. Ultimately, the authors have rewritten the final version of the manuscript in their own words.

**Prompts for Generative Model Evaluation**

You are a highly capable multimodal AI assistant tasked with evaluating answers to visual questions. Please analyze the following image and question, then determine which of the two provided answers is better.

Question: Which caption describes the image better?

Answer 1: [reference or candidate caption]

Answer 2: [reference or candidate caption]

Please evaluate both answers based on the following criteria:
1. Accuracy: How well does the answer align with the visual information in the image?
2. Completeness: Does the answer fully address all aspects of the question?
3. Clarity: Is the answer easy to understand and well-articulated?
4. Relevance: Does the answer directly relate to the question and the image?

After your evaluation, please:
1. Explain your reasoning for each criterion.
2. Provide an overall judgment on which answer is better (Answer 1 or Answer 2). For example: Overall Judgment: Answer X is better.

Your response should be structured and detailed, demonstrating your understanding of both the visual and textual elements of the task.

- - - - - - - - - - - - - - - - - - - - - - - - - - - - - - - - - - - - - - - - - - - - - - - - -

You are a highly capable multimodal AI assistant tasked with evaluating the quality of a caption to the image. Please analyze the following image and caption, then determine the score for the caption in the range of 0.0 (bad quality) to 1.0 (good quality).

Caption: [candidate caption]

Please evaluate the caption based on the following criteria:
1. Accuracy: How well does the caption align with the visual information in the image?
2. Completeness: Does the caption fully address all aspects of the question?
3. Clarity: Is the caption easy to understand and well-articulated?
4. Relevance: Does the caption directly relate to the question and the image?

After your evaluation, please:
1. Explain your reasoning for each criterion.
2. Provide an overall judgment score. For example: Overall Judgment: X.

Your response should be structured and detailed, demonstrating your understanding of both the visual and textual elements of the task.

Table 6: **Prompts for evaluating Molmo-7B, Qwen2-VL-7B, and InternVL2-8B for pairwise (above) and pointwise (below) ranking**. For pairwise ranking evaluation, the model is required to indicate which of the two texts better matches the image. For pointwise ranking, the model must assign a score between 0 and 1 that reflects the quality of the match. We barely modify the original prompts used to evaluate generative reward models in VL-Reward-Bench for pairwise ranking evaluation.

**Prompts for Generative Model Evaluation**

Given an image, please serve as an unbiased and fair judge to evaluate the quality of the captions provided by a Large Multimodal Model (LMM). Determine which caption is better and explain your reasoning with specific details. Your task is provided as follows:
The first caption: [reference or candidate caption]
The second caption: [reference or candidate caption]
ASSISTANT:

- - - - - - - - - - - - - - - - - - - - - - - - - - - - - - - - - - - - - - - - - - - - - - -

Given an image and a corresponding question, please serve as an unbiased and fair judge to evaluate the quality of answer answers provided by a Large Multimodal Model (LMM). Score the response out of 100 and explain your reasoning with specific details. Your task is provided as follows:
Question: [What this image presents?]
The LMM response: [candidate caption]
ASSISTANT:

Table 7: **Prompts for evaluating LLaVA-Critic-7B for pairwise (above) and pointwise (below) ranking**. In the case of evaluating LLaVA-Critic-7B, the pointwise ranking evaluation requires the model to return the actual score within the range of 0 to 100. We notice this model particularly outputs better scores when prompted with the scale of 0 to 100 than 0 to 1, different from the other generative models.

**Prompts for Generative Model Evaluation**

Select which of the captions describes the image better.
Caption 1: [reference or candidate caption].
Caption 2: [reference or candidate caption].
Please either only select integer 1 or 2. Do not include any text-based captions, reasons or punctuation.

- - - - - - - - - - - - - - - - - - - - - - - - - - - - - - - - - - - - - - - - - - - - - - -

**v1**: Rate the following caption for the given image.
Caption: [candidate caption].
Please only provide a rating in the range of 0 (poor quality) to 100 (good quality). Do not include any reasons.

**v2**: Rate the following caption for the given image in terms of how much the caption accurately depicts the image.
Caption: [candidate caption].
Please only provide an integer score from 0 to 100. Do not include any text-based captions, reasons, or punctuation.

Table 8: **Prompts for evaluating LLaMA-3.2-11B for pairwise (above) and pointwise (below) ranking**. Unlike the other generative models, we explicitly instruct the model to exclude the reasons or any text-based captions due to their inconsistencies in response formatting. Despite this constraint, the model achieves the highest preference accuracies among the generative models, as shown in Table 3.

**Prompts for Scalar-based Model Evaluation**

```
{"role": "user",
 "content": 'Describe the image.'}
{"role": "assistant",
 "content": batch_response}
```
- - - - - - - - - - - - - - - - - - - - - - - - - - - - - - - - - - - - - - - - - - - - - - -
```
{"role": "user",
 "content": 'I want to generate the caption from the input image.'}
{"role": "assistant",
 "content": batch_response}
```

Table 9: **Two variations of prompts used for evaluating IXCREW-S**. We present two ways to examine the effects of the user prompts on the performance (Table 21).

**Prompts for LVLM Response Refinement**

You are an expert at providing a Blind or Low Vision (BLV) an accurate, helpful description, given an environmental scene (outdoor or indoor) and corresponding to their text-based request. Remember that BLV users cannot see as much as normally-sighted humans, so you must provide detailed but precise information from the image. Enhance the model response by including precise clock directions (options: 9, 10, 11, 12, 1, 2, 3 o'clock), depth levels (in meters or steps), and objects BLV users should avoid or utilize.

Table 10: **System prompt for refining LVLM responses using GPT-4o mini**. We ensure that the model incorporates the perspectives of both normally sighted individuals and Blind or Low Vision (BLV) users. Descriptions should be accurate in direction and depth, while also adhering to safety considerations.

**Prompts for LVLM Response Scoring**

You will be given one sentence of visual caption generated from one image and request. Your task is to rate the generated caption on one metric.

Evaluation Criteria: Score is from 0 to 100 - The generated caption should accurately fulfill the request based on the image. You should penalize captions that include irrelevant details, omit significant elements indicated by the request, or fail to accurately describe the visual content of the image. Assign an integer score from 0 to 100 for the caption based on the following dimensions:

- Direction Accuracy: Are the directions provided (e.g., angle, clock direction, left or right) in the caption accurate and aligned with the image?

- Depth Accuracy: Are the depth levels (e.g., x meters, x steps) in the caption correct and aligned with the image?

- Response Quality: Is the caption free from unnecessary repetition, illogical order, irrelevance to the scenario, and hallucinations?

- Safety and Actionability: Are all the action verbs included in the caption entirely safe and actionable for blind users to fulfill their request?

- Sufficiency: Does the caption provide all the necessary information for blind users to fulfill their request?

- Conciseness: Is the caption concise and free from verbosity?

- Overall: How would you rate the caption overall?

Request: [request]

Generated Caption: [response]

Provide a JSON output with integer scores for the 7 evaluation criteria.

Table 11: **System prompt for generating scores for constructing EYE4ALL training dataset with GPT-4o mini**. We instruct GPT-4o mini to assess the captions generated by one of the 7B models across seven aspects, aligned with the criteria provided to human annotators.

---

**Labeling Guidelines for LVLM Response Evaluation**

**Overview**: This study aims to evaluate the capability of vision-language models in generating deep context to support the mobility of blind or low-vision (BLV) users. As an annotator, your task is to assess and refine the model-generated responses based on the given indoor or outdoor images and scenarios. You will evaluate the provided responses on several criteria and make necessary corrections to ensure accuracy, usability, and relevance. Each scenario consists of the following:
1. Image: an indoor or outdoor environmental scene provided as visual context.
2. Request: a BLV mobility-related request or task from BLV users.
3. Step-by-step description: the vision-language model's response to the request.

**Notes for Refinement**: The deep context responses often follow this format:
1. Scene Description: An overview of the environment, highlighting key landmarks.
2. Distance and Direction to the Goal: Clear directional and distance information to the target.
3. Obstacles to Watch For: Specific obstacles the user should be aware of.
4. Step-by-Step Directions: Detailed instructions for completing the task.
When a response does not follow this format, refine it accordingly. Copy the original response and correct errors, remove unnecessary details, or add missing information.

**Evaluation Criteria**: For each response, you will rate the following aspects on a Likert scale (1 to 5) or a binary scale and provide corrections where necessary:

1. *Direction Accuracy*
- Definition: Are the directions provided (e.g., angle, clock direction, left or right) in the response accurate and aligned with the image?
- Ratings: 1: no accurate info at all, 2: 3 inaccurate info, 3: 2 inaccurate info, 4: 1 inaccurate info, 5: entirely accurate

2. *Depth Accuracy*
- Definition: Are the depth levels (e.g., x meters, x steps) in the response correct and aligned with the image?
- Ratings: 1: no accurate info at all, 2: 3 inaccurate info, 3: 2 inaccurate info, 4: 1 inaccurate info, 5: entirely accurate

3. *Response Quality* (Step-by-Step Order, Relevance, Hallucination and Repeatedness)
- Definition: Is the response free from unnecessary repetition, illogical order, irrelevance to the scenario, and hallucinations?
- Ratings: 0: no (the response is illogically ordered, and includes irrelevant or hallucinated details, or has repetitions), 1: yes

4. *Safety and Actionability*
- Definition: Are the actions in the response safe and actionable for BLV users?
- Ratings: 1: no safe or actionable actions, 2: no safe actions but actionable actions, 3: safe actions but include non-actionable actions (e.g., watch xxx, see xxx, observe xxx, drive xxx, etc.), 4: includes consensual safety actions (e.g., watch out for cars when crossing the street.), 5: entirely safe and actionable

5. *Sufficiency*
- Definition: Does the response provide all the necessary information for BLV users to complete the task?
- Ratings: 1: strongly disagree, 2: disagree, 3: neutral, 4: agree, 5: strongly agree

6. *Conciseness*
- Definition: Is the response concise and free from verbosity?
- Ratings: 1: strongly disagree, 2: disagree, 3: neutral, 4: agree, 5: strongly agree

7. *Overall*
- Definition: Rate the response overall.
- Ratings: 1: not useful at all, 2: needs significant improvement, 3: needs some improvement, 4: valid for BLV users, 5: very useful for BLV users.

Table 12: **Instruction guideline for human experimental procedure**. We provide an overview and detailed steps for the annotation and refinement processes. The human annotators were also encouraged to ask clarifying questions to the authors whenever needed.

Figure 4: **Sample screenshot of interface used in the human experiment**. This annotation screen with a different image-request-response is shown 100 times per annotator.

**Request:** Enter the Coffee Shop located on the right side of the side walk.

**Response:**
1) Scene Description: You are walking down the sidewalk in front of Cheonggye Plaza, located in the 6 o'clock direction, about 15 meters away.
2) Main Obstacles: Be cautious of vehicles, pedestrians, and street furniture.
3) Step-by-Step Actions
- Step 1: Turn left in the 1 o'clock direction until you reach the curb.
- Step 2: Walk along the sidewalk for 3–5 steps and turn right in the 2 o'clock direction.
- Step 3: Continue 5–7 steps straight to the Starbucks Coffee entrance.

| Conciseness | Sufficiency | Safeness | Hallucination | Direction Acc | Depth Acc | Overall Quality |
|---|---|---|---|---|---|---|
| 0.58 | 0.75 | 0.83 | 0.67 | 0.42 | 0.58 | 0.67 |

Figure 5: **Example of `EYE4ALLMulti`.** `EYE4ALLMulti` comprises a text request, an image, model-generated responses, and scores across seven dimensions: Conciseness, Sufficiency, Safeness, Hallucination, Direction Accuracy, Depth Accuracy, and Overall Quality. These scores are normalized and averaged over 2–3 human annotators.

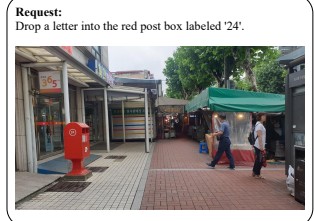

**Request:**
Drop a letter into the red post box labeled '24'.

**Rejected Response (Model generated):**
1) Scene Description: You are near a red post box labeled '24', located on the sidewalk in the 3 o'clock direction, about 15 meters away.
2) Main Obstacles: Be cautious of the vehicle parked on the side of the road and pedestrians walking on the sidewalk.
3) Step-by-Step Actions:
- Step 1: Walk in the 11 o'clock direction until you reach a parked vehicle on the left side of the street.
- Step 2: Turn slightly left in the 2 o'clock direction and walk 10 steps straight.
- Step 3: Continue walking straight in the 3 o'clock direction until you reach the red post box on the sidewalk.

**Preferred Response (Human-refined):**
1) Scene Description: You are near a red post box labeled '24', located on the sidewalk in the 10 o'clock direction, approximately 5 meters ahead.
2) Main Obstacles: Watch for pedestrians walking along the sidewalk and any nearby parked vehicles.
3) Step-by-Step Actions:
- Step 1: Walk straight for 5~6 steps in the 10 o'clock direction.
- Step 2: Stop in front of the red post box and drop the letter into it.

Figure 6: **Example of `EYE4ALLPref`.** `EYE4ALLPref` consists of a text request, an image, model-generated responses, and human-refined responses. For evaluation, model-generated responses are treated as rejected, while human-refined responses are considered preferred samples.

| | PASCAL-50S P-Acc | FOILR1 P-Acc | FOILR4 P-Acc | FlickrExp $\tau_c$ | FlickrCF $\tau_b$ | Polaris $\tau_c$ | Polaris* P-Acc | OID* P-Acc | ImgREW P-Acc |
|---|---|---|---|---|---|---|---|---|---|
| **MULTI-TAP** | | | | | | | | | |
| - Qwen-2B-S | 84.7 | 97.3 | 97.3 | 57.8 | 39.4 | 59.8 | 83.3 | 58.5 | 59.9 |
| - Qwen-7B-S | 83.8 | 97.0 | 97.0 | 56.3 | 39.9 | 61.5 | 84.7 | 57.3 | 60.3 |
| - InternLM-7B-S | 82.0 | 97.1 | 97.1 | 53.1 | 38.8 | 57.3 | 83.7 | 58.9 | 54.3 |
| - LLaMA-3.2-S | 82.7 | 96.5 | 96.5 | 56.9 | 38.3 | 60.9 | 81.3 | 56.5 | 63.5 |

Table 13: **Performances of our MULTI-TAP trained with the empty prompt setting on 8 human preference judgment datasets.** The performances are similar to the main results in Table 1.

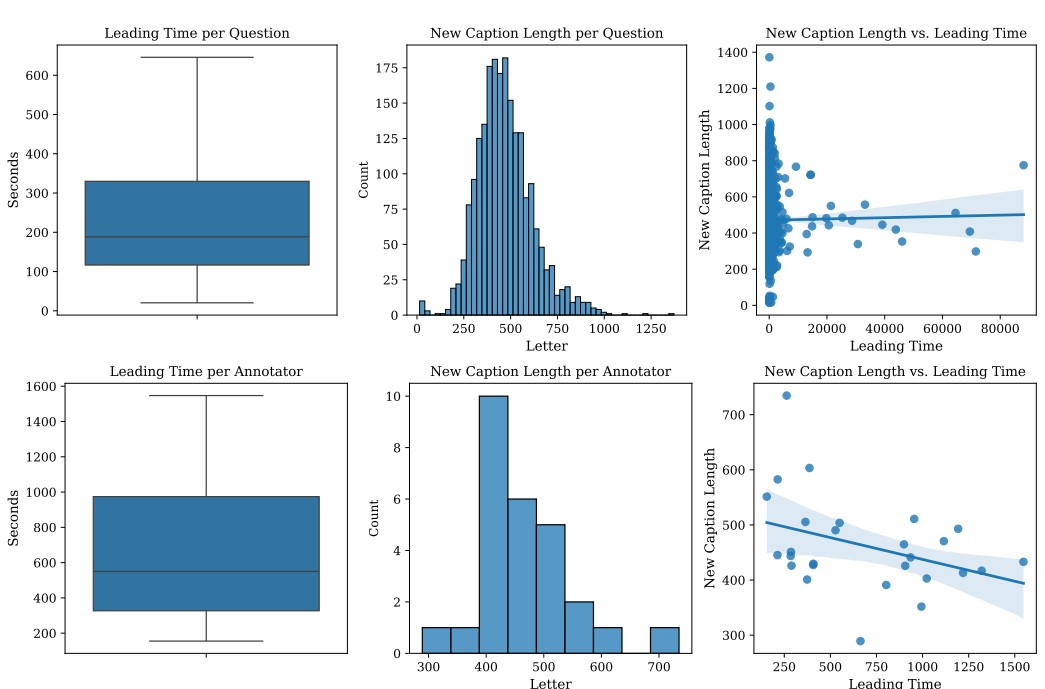

Figure 7: **Distribution and correlation of leading time per question/annotator and the length of newly human-generated captions**. Longer leading time does not necessarily mean more captions generated by human annotators.

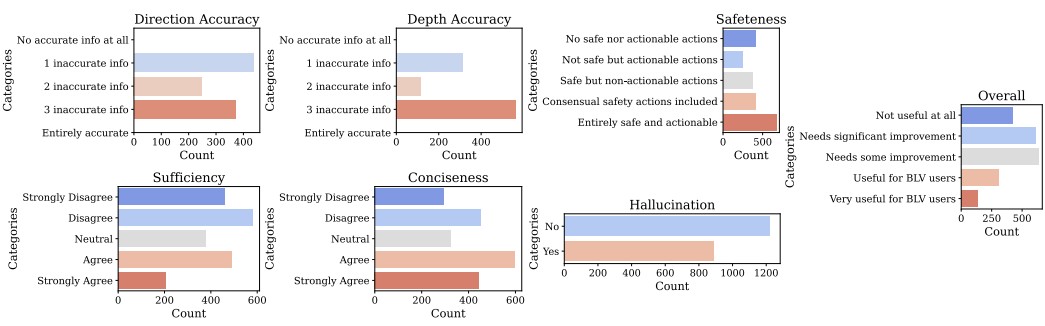

Figure 8: **The distribution of human-annotated scores across seven categories**. These plots indicate a room for significant improvement in LVLMs, especially in terms of accuracy, safety, and sufficiency.

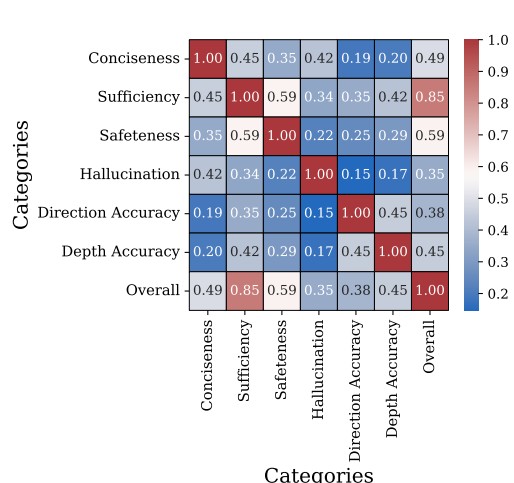

Figure 9: **Correlation of human judgment scores across different category pairs**. The overall quality of LVLM responses is highly correlated with the sufficiency category, captured with Pearson's correlation coefficient of 0.85.

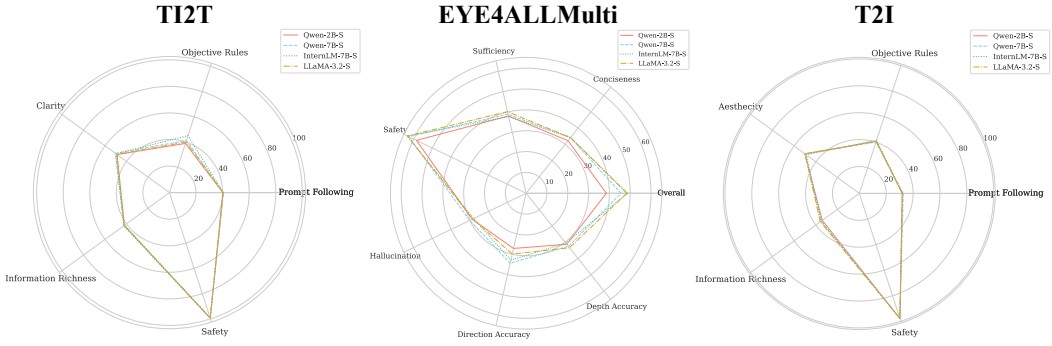

Figure 10: **Performances of our MULTI-TAP models on multi-objective scoring datasets**. The MULTI-TAP models trained on different LVLM architectures show a similar trend across multiple categories for all three datasets.

| Dimension | VisionREW-S | MULTI-TAP$_{Qwen-2B-S}$ | MULTI-TAP$_{Qwen-7B-S}$ | MULTI-TAP$_{InternLM-7B-S}$ | MULTI-TAP$_{LLaMA-3.2-S}$ |
|---|---|---|---|---|---|
| 0 | 90.38 | 91.85 | 90.87 | 91.76 | 90.78 |
| 1 | 1.08 | 99.41 | 99.12 | 99.41 | 98.92 |
| 2 | 94.70 | 97.25 | 96.47 | 97.25 | 96.17 |
| 3 | 10.50 | 94.80 | 94.01 | 94.80 | 93.92 |
| 4 | 26.20 | 75.76 | 75.17 | 73.21 | 74.88 |
| 5 | 10.79 | 93.72 | 92.35 | 93.33 | 92.84 |
| 6 | 99.90 | 99.80 | 99.51 | 99.80 | 99.71 |
| 7 | 93.62 | 93.62 | 92.44 | 93.52 | 92.54 |
| 8 | 11.09 | 90.19 | 89.50 | 90.19 | 89.89 |
| 9 | 1.77 | 99.41 | 98.92 | 99.41 | 99.12 |
| 10 | 99.12 | 99.51 | 99.21 | 99.51 | 99.21 |
| 11 | 92.54 | 95.49 | 94.80 | 95.49 | 94.31 |
| 12 | 14.23 | 89.21 | 88.13 | 88.42 | 87.44 |
| 13 | 98.14 | 96.86 | 96.57 | 96.86 | 96.37 |
| 14 | 78.90 | 73.01 | 69.09 | 70.26 | 70.17 |
| 15 | 7.75 | 89.99 | 89.89 | 89.79 | 89.40 |
| 16 | 0.39 | 99.41 | 99.61 | 99.41 | 99.61 |
| 17 | 93.52 | 95.68 | 94.41 | 95.39 | 94.60 |
| 18 | 11.48 | 91.56 | 90.97 | 91.36 | 90.38 |
| 19 | 92.15 | 95.98 | 95.29 | 95.39 | 94.70 |
| 20 | 5.50 | 94.11 | 93.82 | 94.01 | 93.42 |
| 21 | 98.72 | 99.61 | 99.31 | 99.61 | 98.92 |
| 22 | 74.39 | 81.55 | 79.39 | 79.29 | 79.29 |
| 23 | 9.22 | 85.67 | 84.69 | 85.38 | 84.89 |
| 24 | 99.61 | 99.80 | 99.61 | 99.80 | 99.41 |
| 25 | 89.01 | 92.74 | 91.66 | 92.64 | 91.76 |
| 26 | 99.90 | 100.00 | 100.00 | 100.00 | 100.00 |
| 27 | 94.80 | 92.93 | 92.44 | 92.93 | 92.15 |
| 28 | 7.46 | 92.35 | 91.36 | 92.25 | 91.46 |
| 29 | 0.29 | 99.21 | 99.21 | 99.21 | 99.02 |
| 30 | 99.41 | 99.12 | 98.72 | 99.12 | 98.63 |
| 31 | 89.50 | 90.19 | 89.30 | 89.89 | 88.52 |
| 32 | 34.54 | 62.90 | 61.63 | 61.04 | 62.90 |
| 33 | 7.75 | 90.87 | 90.28 | 90.97 | 90.28 |
| 34 | 2.75 | 97.94 | 97.35 | 97.94 | 97.84 |
| 35 | 0.20 | 100.00 | 99.80 | 100.00 | 99.80 |
| 36 | 94.50 | 93.72 | 93.03 | 93.72 | 92.84 |
| 37 | 41.41 | 55.94 | 56.33 | 54.47 | 53.29 |
| 38 | 8.93 | 90.68 | 89.79 | 90.78 | 89.79 |
| 39 | 0.98 | 98.72 | 98.72 | 98.72 | 98.33 |
| 40 | 97.35 | 93.33 | 92.74 | 93.33 | 92.54 |
| 41 | 69.58 | 68.99 | 66.34 | 65.55 | 65.55 |
| 42 | 54.47 | 53.97 | 54.96 | 50.83 | 52.99 |
| 43 | 46.91 | 53.97 | 53.09 | 51.62 | 53.58 |
| 44 | 45.44 | 55.45 | 52.89 | 51.72 | 54.37 |
| 45 | 99.71 | 99.41 | 99.21 | 99.41 | 99.21 |
| 46 | 96.57 | 94.80 | 94.21 | 94.80 | 94.21 |
| 47 | 81.06 | 73.80 | 72.13 | 72.72 | 71.74 |
| 48 | 59.27 | 55.62 | 55.64 | 55.94 | 56.33 |
| 49 | 54.96 | 53.97 | 53.48 | 53.88 | 56.43 |
| 50 | 99.90 | 99.71 | 99.61 | 99.71 | 99.51 |
| 51 | 96.66 | 96.57 | 96.07 | 96.57 | 96.07 |
| 52 | 89.30 | 87.83 | 86.85 | 87.63 | 87.14 |
| 53 | 73.21 | 71.34 | 69.87 | 68.79 | 68.89 |
| 54 | 70.36 | 68.20 | 66.44 | 66.54 | 65.85 |
| 55 | 13.35 | 93.03 | 91.85 | 92.93 | 91.95 |
| 56 | 7.16 | 94.21 | 93.42 | 94.21 | 93.62 |
| 57 | 3.93 | 96.57 | 96.07 | 99.67 | 96.07 |
| 58 | 0.20 | 99.80 | 99.80 | 99.80 | 99.71 |
| **Avg** | 53.30 | 88.00 | 87.30 | 87.40 | 87.20 |
| **Time** | 51 days | 4 hrs | 6 hrs | 6 hrs | 11 hrs |

Table 14: **Multi-objective performances of VisionREW-S and our MULTI-TAP models on VisionREW dataset across 59 dimensions**. MULTI-TAP outperforms VisionREW-S in terms of both efficiency and performance.

| Dimension | VisionREW-S | MULTI-TAPQwen-2B-S | MULTI-TAPQwen-7B-S | MULTI-TAPInternLM-7B-S | MULTI-TAPLLaMA-3.2-S |
|---|---|---|---|---|---|
| Conciseness | 33.55 | 46.54 | 55.59 | 62.30 | 58.89 |
| Sufficiency | 54.21 | 43.45 | 46.33 | 46.96 | 47.50 |
| Safety | 26.20 | 72.95 | 73.70 | 73.80 | 73.80 |
| Hallucination | 53.14 | 42.81 | 45.26 | 46.65 | 47.92 |
| Direction Acc | 55.38 | 42.07 | 43.13 | 44.73 | 45.69 |
| Depth Acc | 51.01 | 41.00 | 47.07 | 48.35 | 44.62 |
| Overall | 59.96 | 39.62 | 42.07 | 41.75 | 42.71 |
| **Avg** | 47.63 | 46.92 | 50.45 | 52.08 | 51.59 |

Table 15: **Multi-objective performances of VisionREW-S and our MULTI-TAP models on `EYE4ALLMulti`-Binary dataset across seven dimensions**. MULTI-TAPQwen-7B-S, MULTI-TAPInternLM-7B-S, and MULTI-TAPLLaMA-3.2-S show higher average performance than VisionREW-S on our proposed dataset.

| Dimension | VisionREW-S | MULTI-TAPQwen-2B-S | MULTI-TAPQwen-7B-S | MULTI-TAPInternLM-7B-S | MULTI-TAPLLaMA-3.2-S |
|---|---|---|---|---|---|
| Prompt Following Rate | 41.75 | 59.35 | 59.52 | 56.84 | 60.40 |
| Objective Rules | 15.80 | 84.20 | 84.11 | 84.23 | 83.47 |
| Aestheticity | 19.97 | 79.96 | 80.79 | 80.37 | 81.01 |
| Information Richness | 41.94 | 59.28 | 59.20 | 57.25 | 59.94 |
| Safety | 0.78 | 99.22 | 99.22 | 99.22 | 99.10 |
| **Avg** | 24.05 | 76.40 | 76.57 | 75.58 | 76.78 |

Table 16: **Multi-objective performances of VisionREW-S and our MULTI-TAP models on Align-anything (T2I)-Binary dataset across five dimensions**. MULTI-TAP significantly surpasses VisionREW-S performance on the T2I dataset.

| Dimension | VisionREW-S | MULTI-TAPQwen-2B-S | MULTI-TAPQwen-7B-S | MULTI-TAPInternLM-7B-S | MULTI-TAPLLaMA-3.2-S |
|---|---|---|---|---|---|
| Prompt Following Rate | 8.08 | 91.76 | 91.35 | 91.71 | 91.50 |
| Objective Rules | 4.72 | 94.82 | 94.77 | 95.18 | 95.08 |
| Clarity | 1.40 | 98.65 | 98.55 | 98.60 | 98.60 |
| Information Richness | 13.11 | 86.58 | 86.17 | 86.58 | 86.37 |
| Safety | 0.00 | 100.00 | 99.53 | 100.00 | 99.79 |
| **Avg** | 5.46 | 94.36 | 94.07 | 94.41 | 94.27 |

Table 17: **Multi-objective performances of VisionREW-S and our MULTI-TAP models on Align-anything (TI2T)-Binary dataset across five dimensions**. MULTI-TAP significantly surpasses VisionREW-S performance on the TI2T dataset.

**Generated Samples Using Generative Model**

1. Accuracy:
Answer 1: Inaccurate. The image doesn't show a blond woman in a blue shirt waiting for a ride.
Answer 2: More accurate. It correctly describes a young child in blue goggles in a pool setting.

2. Completeness:
Answer 1: Incomplete. It lacks key details about the child's position or the pool setting.
Answer 2: More complete. It fully describes the child, their attire, and the surrounding environment.

3. Clarity:
Answer 1: Poor. It's vague and doesn't clearly convey what's in the image.
Answer 2: Better. It's concise and easy to understand, directly describing what's visible.

4. Relevance:
Answer 1: Irrelevant. It doesn't address the question about the woman in the blue shirt.
Answer 2: Relevant. It directly answers the question about the child in blue goggles.

Overall Judgment: Answer 2 is better.

It accurately describes the key elements of the image, including the child, their attire, and the pool setting, which directly answers the question about what's in the picture.

- - - - - - - - - - - - - - - - - - - - - - - - - - - - - - - - - - - - - - - - - - - -

I'll evaluate the caption based on the given criteria:

1. Accuracy: The caption accurately describes the child wearing blue goggles and sitting in a float, which aligns with the visual information provided.

2. Completeness: The caption fully addresses the question by mentioning both the child and the goggles/float, providing a comprehensive answer.

3. Clarity: The caption is easy to understand, using simple language to convey the information clearly.

4. Relevance: The caption directly relates to the question and the image, focusing on the key elements described in the visual content.

Overall Judgment: 0.9

The caption scores high in most categories, with perfect accuracy, completeness, and clarity. While it could be slightly more concise, it effectively captures the essential elements of the image. The only minor reason for a point deduction is that it could potentially be even more streamlined, but this doesn't significantly impact its overall quality.

Table 18: **Samples of generated responses from Molmo-7B with the first (above) and second (below) prompts in Table 6**. Molmo-7B produces answers based on the criteria given in the prompt and provides the summarized reason for the overall judgment.

**Generated Samples Using Generative Model**

Evaluation:

1. **Accuracy**:
- **Answer 1**: This answer is accurate as it describes a city bus driving do wn a city street, which is clearly visible in the image.
- **Answer 2**: While the sky is indeed blue and clear in the image, the description doesn't capture the primary aspects of the scene, such as the presence of the bus and other elements like the bank and parked cars.

2. **Completeness**:
- **Answer 1**: This answer covers more parts of the scene. It includes the bus, a critical element, and implicitly communicates the urban setting.
- **Answer 2**: This answer focuses only on the sky and doesn't account for other significant scene elements.

3. **Clarity**:
- **Answer 1**: This answer is straightforward and easily understandable.
- **Answer 2**: It's clear but less descriptive and not as comprehensive.

4. **Relevance**:
- **Answer 1**: Relevant as it directly mentions a city bus and the city street setting, aligning closely with the image.
- **Answer 2**: While relevant to the image, the focus on the sky doesn't capture the main elements of the scene.

Overall Judgment: *Answer 1 is better*.
While both answers are relevant, Answer 1 captures the primary subjects of the image (city bus driving down a street) and provides a more comprehensive description. It is more accurate and addresses more aspects of the scene, making it the better choice for the given question.

- - - - - - - - - - - - - - - - - - - - - - - - - - - - - - - - - - - - - - - - - - - - - - - - - -

Evaluation:

1. Accuracy:
The caption generally aligns well with the visual elements of the image. It accurately captures the sinister and eerie feeling of the portal, the detailed look, and the dark tones. However, it doesn't mention the arch and pipework leading to the portal, nor does it touch on the light leak in the top part of the portal.

2. Completeness:
The caption provides a comprehensive description of the image, capturing many details like the ultra-detailed nature, the warm interior light, and the cinematic shot. However, it misses out on describing the arch and pipework that serve as pathways into the portal.

3. Clarity:
The caption is clear and uses descriptive language to convey the mood and tone of the image effectively. The terms like 'cinematic shot', 'ultra-detailed', and 'photo-realistic' are used to paint a vivid picture of the scene, making it easy for the reader to grasp the image's essence.

4. Relevance:
The caption is very relevant as it directly addresses the nature of the image and uses terminology that reflects the detailed and realistic rendering expected from tools like MCBess.

Overall Judgment:
Given these assessments, the caption is extremely good. It accurately reflects the contents of the image while using vivid, relevant, and descriptive language. However, it could be improved by noting additional observations, such as the arch and pipework leading into the portal, and the top light leak in the portal.

Score:
Overall Judgment: 0.85

Table 19: **Samples of generated responses from InternVL2-8B with the first (above) and second (below) prompts in Table 6**. InternVL2-8B answers based on the criteria given in the prompt and also provides the summarized reason for the overall judgment.

---

**Generated Samples Using Generative Model**

The first caption, "A city bus driving down a city street," is better than the second caption, "the sky is clear and blue." The first caption accurately describes the main subject of the image, which is a city bus driving down a city street. This provides a clear and relevant description of the scene depicted in the image. On the other hand, the second caption, while true, does not capture the primary focus of the image and lacks context about the urban environment and the presence of the bus. Therefore, the first caption is more accurate and informative, making it the better choice.
The better caption: [first].

- - - - - - - - - - - - - - - - - - - - - - - - - - - - - - - - - - - - - - - - - - - - - - - - -

The LMM response provides a detailed description of the image, highlighting several key elements such as "deathly portal to the abyss," "ultra detailed," "warm interior light," "cinematic shot," "photorealistic," "octane render," "high definition," "fine details," "sinister tones," "8 k," and "mcbess mood."

However, the response is somewhat disjointed and lacks coherence. It mentions multiple aspects without clearly connecting them to form a cohesive description of the image. For instance, while it describes the image as having a "deathly portal to the abyss," it also mentions "warm interior light" and "cinematic shot," which are contradictory in terms of atmosphere.

Additionally, some terms like "8 k" and "mcbess mood" are not standard descriptive terms for images and may confuse the reader. The response could be improved by providing a more unified and clear description that captures the overall mood and elements of the image effectively.

Score: 60

Reasons:
- The response includes several relevant details about the image.
- However, it is somewhat disorganized and lacks coherence.
- Some terms are not standard or clear, which might confuse the reader.
- Overall, the response could be more polished and focused.

Table 20: **Samples of generated responses from LLaVA-Critic-7B with the first (above) and second (below) prompts in Table 6**. LLaVA-Critic-7B presents multiple justifications for its selection of the preferred caption and assigns an overall score accordingly.

|  | PASCAL-50S
P-Acc | FlickrExp
$\tau_c$ | FlickrCF
$\tau_b$ | Polaris*
P-Acc | OID*
P-Acc | ImgREW
P-Acc |
|---|---|---|---|---|---|---|
| IXCREW-S-v1 | 73.9 | 21.6 | 25.7 | **89.1** | 56.9 | 53.6 |
| IXCREW-S-v2 | **76.2** | **25.6** | **28.0** | 88.7 | **58.5** | **56.3** |
| IXCREW-S-v3 | 74.2 | 17.0 | 25.7 | 81.9 | 57.5 | 53.6 |

Table 21: **Performances of IXCREW-S using two different prompts in Table 9 and the same prompt as ours (with no content for the user)**. Although version 3 is reported in the main paper to match the prompt setting we use for our models, here, we show that slightly different performances can be achieved with different prompt settings.

|  | FlickrExp
$\tau_c$ | FlickrCF
$\tau_b$ | ImgREW
P-Acc |
|---|---|---|---|
| LLaMA-3.2-11B-v1 | -7.88 | 5.49 | 46.0 |
| LLaMA-3.2-11B-v2 | **5.29** | **9.00** | **51.6** |

Table 22: **Performances LLaMA-3.2-11B using two different prompts in Table 8**. In the main paper, we report version 2, which shows higher generalizability across datasets.

