# OpenReview forum: "Multi-Objective Task-Aware Predictor for Image-Text Alignment"
_ICLR.cc/2026/Conference — Submitted to ICLR 2026_

### Official Review · Reviewer_sS7f · 2025-10-26

**Soundness:** 2
**Presentation:** 2
**Contribution:** 2
**Rating:** 2
**Confidence:** 4

**Summary:**

This paper introduces MULTI-TAP — a scalable, plug-and-play framework for evaluating image-text alignment using large vision-language models (LVLMs). The paper also introduces EYE4ALL, a novel dataset targeting blind and low-vision contexts, containing both pairwise human preferences (EYE4ALLPref) and fine-grained multi-dimensional scores (EYE4ALLMulti).

**Strengths:**

* The proposed method of adding a single prediction head on the LVLM is model-agonistic.
* Results demonstrate consistent improvements in inference time reductions (14h→1h per 1k samples).
* The proposed EYE4ALL dataset fills a gap by emphasizing accessibility and BLV user preferences, representing an important step toward more inclusive evaluation.

**Weaknesses:**

* **Unclear motivation and role of the multi-objective setup.**
  The motivation for introducing the multi-objective learning scheme is not clearly articulated. In the current formulation, the multi-objective branch is completely decoupled from the general score prediction task. It is unclear how optimizing multiple objectives contributes to improving the general score prediction, since there is no explicit interaction between them. Moreover, it is not specified which dataset(s) were used to train the multi-objective components — are they based on *Polaris* and *ImageReward* as well? If so, where do the additional annotation dimensions come from? This lack of clarity makes it difficult to assess the necessity and contribution of the multi-objective design.

* **Unfair comparison with IXC-2.5 and unclear interpretation of human preference alignment.**
  The comparison with IXC-2.5 to argue about human preference misalignment is not meaningful. The human preference scores are computed from a **pointwise** dataset, while IXC-2.5 itself is trained on **pairwise** data. Such a comparison is inherently unfair and may not reflect true preference alignment. Moreover, among models that are designed for pointwise prediction, the proposed method’s improvement appears to be quite minor.


* **Lack of justification for the source of performance gains.**
  The paper does not sufficiently justify whether the reported performance improvements mainly stem from the proposed *single reward head training strategy* or from differences in training data. The baseline models used for comparison appear to be trained on different datasets (i.e. IXCREW-S), which undermines the fairness of the comparison. In particular, since the proposed model has been trained on *Polaris*, it naturally achieves higher Kendall’s correlation on the *Polaris* test set — suggesting that the observed gains may largely reflect data overlap rather than model improvements.

* **Misattribution of long-sequence capability as a method advantage.**
  The paper highlights long-sequence processing as one of the key advantages of the proposed approach. However, this capability appears to derive primarily from the underlying LVLM backbone and its context window length, rather than from the proposed method itself. Therefore, it should not be considered an intrinsic advantage of the method. The paper would benefit from a clearer distinction between the contributions of the proposed training strategy and the inherited properties of the base model.

**Questions:**

* The paper’s organization and figure captions are not sufficiently clear. For instance, in *Figure 1*, the label “Proposed Metric” seems misleading and should likely be “Proposed Method,” as the figure aims to compare proposed method with previous methods.
* In *Figure 2*, the “Single-objective Reward Head” block is not colored, while “Multi-objective Regression” is highlighted. It is unclear whether these color differences convey any specific meaning or are merely stylistic. The figure would benefit from a clearer and more consistent visual convention to avoid reader confusion.

---

> ### Author Response · Authors · 2025-11-17
>
> We sincerely thank the reviewers for their time and thoughtful feedback. We are encouraged that the reviewer found the proposed architecture efficient, model-agnostic, and socially valuable through its accessibility-oriented EYE4ALL dataset. Below, we address each concern point-by-point.
>
> **W1. Unclear motivation and role of the multi-objective setup**
>
> We appreciate the reviewer’s question regarding the motivation and role of the multi-objective branch.  We clarify that the multi-objective branch in MULTI-TAP is intentionally decoupled from the single-objective predictor to ensure modularity, interpretability, and training stability. Rather than jointly optimizing with the overall score, it provides fine-grained, human-interpretable evaluations across dimensions such as accuracy, safety, and sufficiency, each trained on the annotated splits of VisionREW, Align-anything, and EYE4ALLMulti. Hence, we emphasize that optimizing single objectives contributes to shaping semantically rich multimodal embedding (L158) used for evaluating single objective setting and training for multi objective setting, not the other way around.
>
> Empirically, the results of the single-objective learning is demonstrated in Tables 1 to 4, and the results of the multi-objective learning is in Figure 3, Figure 10, and Tables 13—16.  MULTI-TAP outperforms the prior state-of-the-art multi-objective predictor (VisionREW-S), demonstrating the value of this asynchronous design. The decoupled structure prevents gradient interference, supports efficient reuse of multimodal embeddings, and allows lightweight new heads to be added for unseen tasks. This enables the framework scalable, adaptable to diverse evaluation settings, and compatible with evolving human-defined criteria while preserving the integrity of the main scalar predictor.
>
> **W2. Unfair comparison with IXC-2.5 and unclear interpretation of human preference alignment**
>
> We respectfully disagree that our comparison with IXCREW-S (IXC-2.5-Reward) is unfair. Although IXCREW-S is trained in a pairwise fashion, **a reliable reward or evaluation model should preserve consistent rankings when evaluated on pointwise data, as ranking consistency is a fundamental property** in reinforcement learning from human feedback. Measuring Kendall’s τ correlation, therefore, provides a fair and meaningful way to evaluate alignment between model predictions and human judgments, regardless of the supervision format.
>
> Crucially, MULTI-TAP demonstrates robust performance on both pairwise and pointwise evaluations. As shown in Table 1, our model achieves higher accuracy on pairwise datasets such as FOILR1/4 and PASCAL-50S, while also maintaining stronger Kendall’s correlation on pointwise datasets like FlickrExp and Polaris. This consistent superiority across different evaluation types supports that our approach generalizes ranking consistency rather than optimizing for a specific annotation type.
>
> We define human preference alignment as the ability of a model to preserve human-consistent ranking relations across modalities, independent of whether training supervision is pairwise or pointwise. This view aligns with standard practices in reward-modeling research, where relative ranking fidelity is considered the primary criterion for alignment. Taken together, the results indicate that MULTI-TAP achieves a genuine improvement in human preference alignment, not simply a task-specific advantage. We will clarify this definition and interpretation in the revised version to avoid potential misunderstanding.

---

> ### Author Response · Authors · 2025-11-17
>
> **W3. Lack of justification for the source of performance gains**
>
> We appreciate the reviewer’s concern regarding the potential influence of training data overlap and would like to clarify that the observed performance gains do not stem from data redundancy. While MULTI-TAP was trained on Polaris and ImageReward, it was extensively evaluated on a diverse set of heterogeneous and unseen benchmarks, including FlickrExp, FOILR, PASCAL-50S, Sightation, Align-anything, EYE4ALLPref, and EYE4ALLMulti. Also, we have used only the training split of both Polaris and ImageReward for training like Polos and ImgREW-S, respectively. The consistent improvements across these out-of-distribution datasets demonstrate that our approach generalizes well beyond the training distributions, indicating that the gains primarily arise from the proposed scalar reward-head formulation and training framework rather than from dataset overlap.
>
> We also note that the training data of IXCREW-S are not publicly disclosed, which makes exact replication impossible. To ensure fairness, we adopted its officially released open-source weights and followed the evaluation procedures. MULTI-TAP achieves superior and more consistent performance than IXCREW-S across both pointwise and pairwise benchmarks, underscoring the robustness and reliability of our design.
>
> On a final note, our framework is backbone-agnostic and can be seamlessly instantiated on various LVLMs without architecture-specific constraints. It also supports both single- and multi-objective scoring under a unified paradigm, offering interpretability and scalability that prior scalar-based predictors lack. This methodological generality, combined with cross-benchmark generalization and significant inference efficiency, illustrates that the observed improvements reflect genuine advances in model design rather than differences in training data. We will emphasize this clarification and the advantages of our framework in the revised version.
>
> **W4. Misattribution of long-sequence capability as a method advantage**
>
> We would like to clarify that our paper already acknowledges this distinction. As explicitly stated in L183–185, the ability to handle long sequences originates from the LVLM backbone itself, not from our proposed method. We did not claim this capability as an intrinsic advantage of MULTI-TAP, but rather recognized it as an inherited property of the underlying LVLM architecture. Rather, the long-sequence processing was one of the four criteria that we considered it as an important motivation when building a robust multimodal metric.
>
> Our contribution lies in effectively utilizing this property within a scalar-based evaluation framework, demonstrating that long-context understanding can be leveraged for efficient and direct scoring without the need for generative inference or post-processing. While LVLMs inherently possess this capability, current multimodal evaluation research still relies heavily on generative-based scoring methods or encoder-based metrics such as CLIP-S. Our work expands this landscape by showing that a scalar-based LVLM predictor can achieve comparable semantic alignment with far greater efficiency and interpretability.
>
> Hence, the focus of our study is not on claiming longer context windows as a method-level innovation, but on illustrating how such architectural strengths can be integrated into a lightweight, generalizable evaluation framework. We will make this distinction clearer in the revised version to prevent potential misinterpretation of our claims.
>
> **Q1-2. Figure clarity and labeling**
>
> We thank the reviewer for this keen observation. We will revise Figure 1 by changing “Proposed Metric” to “Proposed Method” for accuracy. In Figure 2, we will unify the color scheme (gray for shared LVLM components, blue for learnable modules) and explicitly explain the legend in the caption to ensure consistent semantics and visual clarity.
>
> We sincerely appreciate the reviewer’s detailed feedback, which helped us strengthen the clarity and transparency of our work. All the above clarifications and figure improvements will be incorporated into the revised version. We would truly appreciate it if you could reconsider the score if the points have been clarified. Thank you again for your time in providing constructive comments.

---

> ### Comment · Reviewer_sS7f · 2025-11-25
> **Response to the Rebuttal**
>
> To Authors,
>
> I have read the rebuttal. My main concern still remains unresolved, about the role and contribution of the multi-objective branch. The fact is  most core experiments and claims rely essentially on the scalar head’s scores. As a result, the multi-objective component feels more like an auxiliary add-on than a practically impactful part of the method, yet it is presented as a key selling point of the paper. There is no clear evidence that it meaningfully affects or improves the main evaluation metric in real use.
>
> Given this, along with my earlier concerns about the strength of the empirical comparisons, I do not find the contribution sufficiently compelling for acceptance at this venue, and therefore I will not change my rating.

---

> ### Author Response · Authors · 2025-11-25
>
> Thank you for the follow-up comment. We would like to emphasize that the multi-objective branch is *not* an auxiliary component but a core contributor to MULTI-TAP’s practical usability. **In real evaluation scenarios, particularly in accessibility and safety-sensitive contexts, the ability to predict both an overall score and fine-grained scores across diverse human criteria (accuracy, safety, sufficiency, etc.) is essential and cannot be achieved with scalar-only predictors.**
>
> On top of that, unlike VisionReward, **our multi-objective design is computationally lightweight and significantly reduces inference cost**, since each objective is predicted by a simple regression head without additional generative decoding (like traditional generative models). This makes MULTI-TAP more scalable and more deployable in large-scale evaluation pipelines.
>
> Contrary to your comment, our work provides **clear evidence of meaningful impact on real-world evaluation metrics**, as demonstrated in Figures 3 and 10 and Tables 13–16. Our newly released Eye4ALL dataset includes human-validated judgment scores across seven interpretable dimensions, all of which necessitate evaluation through our multi-objective predictor.
>
> Combined with its consistent improvements over the previous state-of-the-art multi-dimensional evaluator (VisionREW-S), we strongly believe that the multi-objective branch is a central, practically impactful component. We will make this role clearer in the revised version. We sincerely hope that, given these clarifications, you might reconsider whether the contribution of the multi-objective component may be more substantial than initially interpreted.
>
> Thank you for your time in reviewing our work.

---

### Official Review · Reviewer_mKHf · 2025-10-29

**Soundness:** 3
**Presentation:** 3
**Contribution:** 3
**Rating:** 6
**Confidence:** 3

**Summary:**

This paper presents MULTI-TAP, a multi-objective, task-aware predictor for evaluating image–text alignment. It builds a scalar reward head on top of large vision language models (LVLMs) for single-objective alignment and adds a ridge regression layer for multi-objective scoring. The authors also introduce EYE4ALL, a dataset featuring blind and low-vision user preferences. Experiments show MULTI-TAP achieves higher correlation with human judgments and greater efficiency than prior models such as CLIP-S, IXCREW-S, and VisionREW-S.

**Strengths:**

**Comprehensive formulation**: The paper clearly defines the limitations of existing image–text evaluation models and proposes a unified multi-objective predictor that can handle long-sequence inputs and multi-dimensional alignment.

**Architecture simplicity and scalability**: The two-stage design—frozen LVLM features with ridge regression—balances interpretability, modularity, and low computational cost.

**New dataset (EYE4ALL)**: The dataset broadens evaluation to accessibility-critical scenarios with BLV-oriented annotations, representing a valuable contribution to human-centered multimodal evaluation.

**Strong experimental results**: MULTI-TAP consistently outperforms existing metrics across a variety of datasets and model backbones, showing robustness and cross-architecture generalization.

**Weaknesses:**

**Limited theoretical novelty.** The proposed two-stage design—adding a scalar reward head in Stage 1 and fitting a linear ridge regression on frozen LVLM embeddings in Stage 2—is conceptually simple and builds upon existing practices in feature reuse and reward modeling. While effective, it lacks theoretical or algorithmic innovation beyond a well-engineered combination of known components.

**Insufficient ablations**: The multi-objective head searches α only by training loss and trains one epoch per α, which risks selection bias and underexploration of generalization.

The interaction between Stage 1 (single-objective) and Stage 2 (multi-objective) components is not explored; the paper does not show how each stage contributes to the final performance.

**Questions:**

1.Could you elaborate on why a linear ridge regression was chosen for multi-objective prediction? Have you tested other mappings (e.g., MLP or attention-based fusion) to verify whether the simplicity of ridge regression limits performance or interpretability?

2.MULTI-TAP underperforms IXCREW-S in the TI2T setting. Can you explain what causes this degradation.

3.Several datasets were reformulated or subsampled (e.g., OID*, Polaris*). Could you clarify how these modifications influence the comparability of results with prior works, and whether original test splits were also evaluated?

4.Other questions refer to weakness

I will consider increasing the score based on the authors’ response.

---

> ### Author Response · Authors · 2025-11-19
>
> We sincerely thank the reviewer for their thoughtful and constructive feedback. We deeply appreciate the recognition of our comprehensive formulation, the clarity of the two-stage architecture, the accessibility-centered EYE4ALL dataset, and the robustness of our experimental results across LVLM backbones and benchmarks. We address the raised concerns and questions in detail below.
>
> **W1. Limited theoretical novelty**
>
> We understand the reviewer’s point regarding the conceptual simplicity of the proposed two-stage design. Our primary motivation was not to introduce a new theoretical paradigm but to revisit and address a gap in the current multimodal evaluation landscape, where scalar-based LVLM predictors remain largely under-explored despite their potential efficiency and interpretability.
>
> Most existing reward or evaluation models rely on generative-based predictors, such as LLaVA-Critic, or on encoder-based metrics, such as CLIP-S. We find that generative approaches are computationally expensive for large-scale inference, while encoder-based metrics are unable to capture long-sequence dependencies. By contrast, our framework demonstrates that scalar-based LVLM predictors can achieve high performance with minimal computational cost while remaining backbone-agnostic and easy to deploy across various architectures.
>
> Although the two-stage design builds upon existing learning paradigms, its contribution lies in reformulating the reward modeling process into a modular, interpretable, and efficient framework that unifies single- and multi-objective evaluation. We also contribute a novel BLV-oriented dataset that broadens the scope of multimodal evaluation beyond conventional domains. We release all the data and code to ensure transparency. In this sense, the work provides methodological and practical values toward the development of fair, accessible, and barrier-free AI systems.
>
> **W2. Insufficient ablations and unexplored interaction between stages**
>
> Empirically, we observed rapid and stable convergence: the loss plateaued well before the end of the first epoch, and additional passes over the data produced negligible changes in both the learned coefficients and downstream performance (87.2, 94.07, and 75.58 on EYE4ALLMulti, TI2T, and T2I). Given this consistent behavior across datasets, combined with the need to sweep multiple α values, training beyond a single epoch would substantially increase compute costs without meaningful performance gains. Thus, a single epoch setting provides an efficient yet empirically sufficient solution for this work, and exploring whether longer schedules yield measurable improvements is left for future study.
>
> Regarding the interaction between Stage 1 and Stage 2, we designed the framework such that Stage 1 provides stable and generalizable multimodal embeddings, while Stage 2 operates independently for aspect-specific scoring. The two stages serve complementary purposes: Stage 1 establishes human-aligned representations, and Stage 2 converts them into interpretable fine-grained metrics, where the results are shown in Tables 1 to 4, and Figure 3, respectively, in the main paper (Stage 1 ablation results: Table 12, 20, and 21, and Stage 2 additional results: Figure 10, Table 13-16 in Apppendix). Our intention was to emphasize the modularity and extensibility of this design rather than inter-stage co-adaptation. We will provide more clarity on this rationale and explanation in the revision.

---

> ### Author Response · Authors · 2025-11-19
>
> **Q1. Choice of linear ridge regression for multi-objective prediction**
>
> We sincerely thank the reviewer for this thoughtful question. Our choice of ridge regression was guided by the goal of maintaining simplicity, interpretability, and computational efficiency in the multi-objective module. As described in L455–457, even this lightweight linear mapping achieves strong performance, with an average accuracy of 87.2% on EYE4ALLMulti and 94.07% on TI2T-Binary, both of which far exceed the 53.3% average reported for VisionREW-S.
>
> To further examine whether the simplicity of ridge regression limits model performance, we conducted additional experiments using small MLPs and Random Forests under identical training configurations on the VisionREW benchmark. The results are summarized below:
>
> | **Model Type** | **Average Accuracy (%)** | **Inference Complexity** |
> | --- | --- | --- |
> | **Ridge Regression (ours)** | 87.2 | O(d), d: hidden dimension |
> | MLP | **88.6** | O(d*100+100^2) |
> | Random Forest | 88.2 | O(T*D), T(=100): # trees, D(=10), maximum tree depth |
>
> These results demonstrate that nonlinear mappings provide only marginal accuracy gains while introducing higher computational overhead and reduced stability. Importantly, on the VisionREW benchmark, which includes 59 fine-grained evaluation dimensions, ridge regression still achieves strong and consistent results. This finding indicates that the multimodal embeddings learned in Stage 1 already encode sufficiently rich structure to enable accurate multi-objective prediction without the need for complex nonlinear transformations.
>
> We therefore adopted ridge regression to highlight that a lightweight, interpretable, and efficient model can deliver competitive accuracy even under high-dimensional, multi-objective conditions. This choice aligns with our design philosophy of building a scalable and plug-and-play evaluation framework that generalizes effectively across diverse LVLM backbones. We will clarify this rationale and include these ablation results in the revised manuscript.
>
> **Q2. MULTI-TAP underperformance in the TI2T setting**
>
> We appreciate this thoughtful question. The relative degradation observed in the TI2T setting primarily arises from the training data composition and modality coverage. MULTI-TAP was trained on Polaris (I2T-style) and ImageReward (T2I-style), which both provide rich alignment scores but lack explicit supervision for text–image–to–text (TI2T) reasoning. These datasets are optimized for direct image–text correspondence rather than conditional text generation or interpretation given both modalities. We also considered datasets such as Align-anything, which include multi-dimensional annotations and richer interaction signals, but these were ultimately reserved for multi-objective evaluation rather than training.
>
> The scarcity of instruction-conditioned multimodal datasets remains a limitation in current open resources. While our EYE4ALL dataset begins to address this gap, the field still lacks large-scale data that directly couples image and textual requests with corresponding textual responses in a TI2T format. We therefore view this as an important direction for future work. Expanding training to include instruction-enriched or mixed-modality pre-training could further improve generalization to TI2T tasks, though this would require substantial human annotation efforts.
>
> Despite this constraint, MULTI-TAP still achieves competitive performance across other evaluation settings and demonstrates strong cross-architecture robustness. We will clarify this limitation and discuss potential extensions for TI2T adaptation in the revised manuscript.

---

> ### Author Response · Authors · 2025-11-19
>
> **Q3. Dataset reformulation and comparability**
>
> We acknowledge that more clarification is needed on the dataset reformulation. Note that we have reported the results for the test split of the original dataset - Polaris in the last column of the results in Table 1, 3, 12, and 20. This dataset contains human-rated scores corresponding to how well the captions are written given the image. To create the binary format (Polaris⁎), we selected captions scoring below 0.5 as negative samples and treated the highest-scoring ground-truth captions as positives to account for pairwise ranking evaluation.
>
> For OID, we used the official test split of version 2. Here, each image contains two captions with an average user rating; hence, we selected samples that included both high (≥0.5) and low (<0.5) ratings to construct positive and negative pairs. These transformations follow established practices in prior reward-modeling work and preserve fair comparability. We will clarify these preprocessing steps in the revised version to ensure full transparency.
>
> We sincerely thank the reviewer again for their thoughtful comments and appreciation of our efforts. Their feedback has been invaluable in clarifying the practical and methodological contributions of our work. We will revise the paper to better articulate the motivation behind our scalar-based approach, elaborate on dataset details, and highlight future directions suggested by the reviewer.

---

> ### Author Response · Authors · 2025-11-27
> **Gentle Reminder**
>
> Dear Reviewer mKHf,
>
> With about one week remaining in the review period, we wanted to follow up briefly and thank you again for your thoughtful and constructive feedback. We hope that our responses and additional experiments have addressed your concerns. If there is any further clarification or detail that would support your assessment, we would be pleased to provide it.
>
> Kind regards,
>
> The Authors

---

> > ### Comment · Reviewer_mKHf · 2025-11-28
> >
> > Thank you for the detailed rebuttal and revisions. The responses address my concerns, and I will keep my origin score.

---

> ### Author Response · Authors · 2025-11-28
>
> Dear Reviewer,
>
> Thank you for your thoughtful feedback and for noting that our rebuttal addressed your concerns.
>
> If there are any remaining points we could further clarify or improve to support a higher evaluation, we would be grateful for your guidance. We are happy to provide additional details or revisions as needed.
>
> Thank you again for your time and consideration!

---

### Official Review · Reviewer_yUqU · 2025-10-30

**Soundness:** 3
**Presentation:** 3
**Contribution:** 3
**Rating:** 6
**Confidence:** 3

**Summary:**

This paper introduces MULTI-TAP (Multi-Objective Task-Aware Predictor), a plug-and-play scalar model for evaluating image–text alignment. Built on large vision–language models, MULTI-TAP first learns a single-objective reward head aligned with human judgments using MSE loss, then adds a lightweight ridge regression layer to generate fine-grained multi-objective scores. The authors also present EYE4ALL, a new dataset capturing human and blind/low-vision preferences across seven evaluation dimensions. Experiments show that MULTI-TAP achieves stronger correlation with human ratings and higher efficiency than existing metrics (e.g., CLIP-S, IXCREW-S), performs competitively with GPT-4o-based G-VEval, and surpasses VisionREW-S on multi-objective evaluation tasks.

**Strengths:**

**Simple, pragmatic design:** The two-stage paradigm  strikes a balance between interpretability, efficiency, and portability across LVLM backbones.

**Multi-objective interpretability:** Dimension-specific continuous scores avoid fragile learned aggregation weights and better support task-aware evaluation and product use cases.

**Efficiency & long-context handling:**  Notable inference-time improvements vs. LLM-as-a-judge and per-dimension reward models; competitive under long-context inputs across I2T/T2I/TI2T.

**Weaknesses:**

**Limited ablations/attribution**:  The claim that **MSE** is superior to Bradley–Terry/pairwise losses lacks a **controlled ablation** under identical settings (same data/backbone/schedule) against BT/logistic pairwise/Huber/rank-based losses.

**Potential data overlap/leakage**: Stage 1 trains on Polaris/ImgREW and evaluates on related/variant benchmarks (e.g., Polaris*). More explicit leakage checks and sensitivity analyses are needed (e.g., overlap statistics).

**Aggregation & calibration**: The paper purposely avoids aggregating multi-objective scores, but many applications require **user/task-specific aggregation. Missing post-hoc calibration and explainable aggregators to ensure cross-dataset comparability.

**Questions:**

1. Please provide controlled comparisons of MSE vs. Bradley–Terry/rank-based  losses (acc, convergence, stability).

2. Why is ridge preferred over small MLPs or low-rank linear heads? Did you experiment with similar architectures, and if so, what differences did you observe in performance or stability?

3. How does MULTI-TAP compare to G-VEval and other generative judges on accuracy–cost trade-offs?

---

> ### Author Response · Authors · 2025-11-19
>
> We sincerely thank the reviewer for their thoughtful and encouraging feedback. We appreciate the positive assessment of our two-stage formulation, the interpretability and efficiency of MULTI-TAP, and the recognition of its practicality across LVLM backbones. Below, we respond to the reviewer’s concerns and questions in detail.
>
> **W1 & Q1. Limited ablations and justification for using MSE over Bradley–Terry or pairwise losses**
>
> We appreciate the reviewer’s insightful comment regarding the choice of MSE over Bradley–Terry (BT) and other pairwise or rank-based losses. We indeed conducted preliminary comparisons on Qwen2-VL-2B, training separate reward heads under identical configurations using BT and MSE losses. In these experiments, the BT-trained model exhibited underfitting and unstable spikes in the training loss. Its performance on the Polaris* benchmark fell below a P-Acc of 65, significantly lower than the MSE-trained counterpart.
>
> We attribute this to the nature of datasets (Polaris and ImageReward), where each sample may involve multiple captions per image or multiple images per caption. Pairwise formulations require substantially more combinations of comparisons and sampling steps, leading to slower and less stable convergence. In contrast, MSE directly aligns the scalar outputs with human scores, providing smoother gradients and more consistent optimization.
>
> Our decision to adopt MSE was therefore empirical and pragmatic, driven by convergence stability and generalization performance. We will include a controlled ablation summary comparing these losses in the revised manuscript to make this reasoning explicit.
>
> **W2. Potential data overlap or leakage**
>
> We fully understand the reviewer’s concern regarding possible overlap between training and evaluation data. Both Polaris and ImageReward were sourced directly from their official HuggingFace releases, which already include distinct train, validation, and test splits. We adhered strictly to these predefined splits during training and evaluation, as is standard practice in prior work using these datasets.
>
> Given this separation, data leakage is highly unlikely. However, we acknowledge that inadvertent overlaps can occasionally occur due to earlier dataset versions or caption reuse across benchmarks. We will perform an additional overlap check and include a sensitivity analysis of potential data redundancy in the revision. We appreciate the reviewer for highlighting this point, as it will help us further strengthen the rigor and transparency of the paper.
>
> **W3. Aggregation and calibration of multi-objective scores**
>
> We sincerely thank the reviewer for this thoughtful comment. Our decision not to perform post-hoc calibration or include an additional aggregation head was deliberate. The core philosophy of MULTI-TAP is to demonstrate that the model can achieve user- and task-agnostic alignment purely through its two-stage formulation, without requiring further calibration or learned weighting across dimensions. In practice, the ability to produce stable and meaningful scores directly from the second stage is one of our model’s key strengths.
>
> As discussed in L224–L226, existing multi-objective predictors such as VisionREW-S often rely on an aggregation head whose learned weights are highly unstable, with several dimensions assigned near-zero coefficients. This instability leads to inconsistent overall scores across datasets and necessitates additional calibration steps. Similarly, ArmoRM[1] introduces task-specific gating layers prior to aggregation to compensate for this variability.
>
> In contrast, MULTI-TAP inherently avoids this issue because its Stage 1 predictor already learns an overall reward score aligned with human judgment. The Stage 2 regression heads then provide interpretable, dimension-specific scores that remain well-calibrated relative to this global signal. As a result, no extra calibration or aggregation is required to maintain consistency or interpretability.
>
> We fully acknowledge that for application-level customization, users may still wish to aggregate or reweight scores based on task-specific needs, and our modular outputs support such flexibility externally. However, within the scope of this study, our goal was to show that effective alignment and interpretability can be achieved without post-hoc calibration, which we view as an important methodological advantage of our approach. We will make this rationale clearer in the revised version.

---

> ### Author Response · Authors · 2025-11-19
>
> **Q2. Choice of ridge regression over MLP or low-rank linear heads**
>
> We thank the reviewer for this insightful question. To examine whether the simplicity of ridge regression limits performance, we conducted additional experiments using **MLP** and **Random Forests** under identical configurations and data splits. As shown below, these nonlinear mappings provided only marginal gains in accuracy while increasing computational cost and training variance.
>
> Our ridge regression head achieves strong and consistent results across benchmarks, including 87.2% average accuracy on VisionREW (59 dimensions) and 94.07% on TI2T-Binary. In comparison, MLPs and Random Forests achieved slightly higher scores (88.6% and 88.2% on VisionREW, and 94.44% and 95.25% on TI2T-Binary, respectively), but at the expense of efficiency and stability. These differences indicate that the multimodal embeddings learned in Stage 1 already encode sufficiently rich structure, making complex nonlinear mappings unnecessary.
>
> | **Model Type** | **Average Accuracy (%)** | **Inference Complexity** |
> | --- | --- | --- |
> | **Ridge Regression (ours)** | 87.2 | O(d), d: hidden dimension |
> | MLP | **88.6** | O(d*100+100^2) |
> | Random Forest | 88.2 | O(T*D), T(=100): # trees, D(=10), maximum tree depth |
>
> Given these minimal performance gaps, we selected ridge regression as the final model for its lightweight computation, strong stability, and high interpretability. Its simplicity allows asynchronous and scalable multi-objective scoring across diverse LVLM backbones without introducing the complexity or instability associated with nonlinear heads. This design choice reflects our broader goal of achieving robust and efficient evaluation while preserving model transparency.
>
> **Q3. Comparison to G-VEval and generative judges on accuracy–cost trade-offs**
>
> We appreciate the reviewer’s question about the trade-off between accuracy and computational efficiency. In our comparison, G-VEval was accessed through an API, while other generative judges and our MULTI-TAP were evaluated using a single-GPU inference setup. The “Time (hrs)” values reported in Table 3 represent the total inference time required to process the entire *Polaris* benchmark. Since G-VEval runs as a proprietary API, its hardware configuration and latency cannot be directly controlled or measured, and therefore, its runtime is not reported.
>
> Under these comparable conditions, MULTI-TAP achieves performance on par with or better than G-VEval while requiring dramatically less inference time. For instance, our 7B model reaches a Kendall’s τ of 59.3 on FlickrExp and a P-Acc of 97.8 on FOILR1, matching the GPT-4o-based G-VEval’s results with only a single GPU and without any API overhead. This demonstrates that MULTI-TAP offers a far more favorable accuracy–cost balance, providing competitive predictive quality at a fraction of the computational expense associated with generative judges.
>
> We once again thank the reviewer for the generous and insightful feedback. Their suggestions will help us clarify our experimental choices, strengthen our discussion of calibration and ablation, and highlight the efficiency–accuracy advantages of our approach. All these clarifications will be incorporated into the revised version.
>
> [1] Wang, Haoxiang, et al. "Interpretable preferences via multi-objective reward modeling and mixture-of-experts." *arXiv preprint arXiv:2406.12845* (2024).

---

> > ### Comment · Reviewer_yUqU · 2025-11-23
> >
> > Thank you for the thorough and constructive review. I appreciate your feedback and will keep my score unchanged.

---

> > > ### Author Response · Authors · 2025-11-25
> > >
> > > Thank you for the time and thoughtful feedback you provided throughout the review process. We appreciate the comments to further strengthen the work. If there are any remaining concerns, please let us know, and we would be glad to provide explanation.

---

### Official Review · Reviewer_hQRF · 2025-10-30

**Soundness:** 3
**Presentation:** 3
**Contribution:** 3
**Rating:** 4
**Confidence:** 4

**Summary:**

This paper introduces MULTI-TAP, a novel plug-and-play architecture designed for efficient and robust evaluation of image-text alignment quality. It adopts a two-stage training strategy built on top of pretrained LVLMs, capable of producing not only an overall scalar score but also multi-dimensional fine-grained scores. MULTI-TAP effectively addresses the limitations of existing metrics in terms of human correlation, long-text handling, reasoning efficiency, and multi-objective scoring. In addition, the authors release a new dataset, EYE4ALL, aimed at evaluating text-image-text alignment for blind and low-vision users. Experimental results demonstrate that MULTI-TAP outperforms existing scalar reward models in both performance and efficiency.

**Strengths:**

(1) The paper's core claims are well supported by solid experiments. The rationality of the two-stage approach is validated across different LVLM backbones (Qwen2-VL, InternLM, LLaMA-3.2). The authors also conduct extensive comparisons with various state-of-the-art predictors, including the GPT-4o-based G-VEval.

(2) The authors additionally contribute the EYE4ALL dataset, filling the gap in TI2T evaluation benchmarks that reflect the preferences of assistive technology and BLV communities, which holds significant practical value.

**Weaknesses:**

(1) The LVLM responses in the EYE4ALL dataset are refined using GPT-4o mini, introducing a dependency on a proprietary model and potentially biasing the dataset toward GPT's output style, which may affect model generalization. Moreover, the dataset requires manual verification to ensure quality.

(2) The second stage employs Ridge Regression to learn multi-objective scores. This linear model may limit the ability to capture highly nonlinear interactions among objectives, and as the number of targets increases, the scores may become less accurate.

**Questions:**

The paper mentions that the single-objective score (Overall Score) in Stage 1 is highly correlated with the fine-grained score (Sufficiency) in Stage 2. Could this correlation be leveraged to guide the training process in Stage 2?

---

> ### Author Response · Authors · 2025-11-18
>
> We sincerely thank the reviewer for the detailed and thoughtful feedback. We are encouraged that the reviewer recognized the strength of our experimental validation, the rationality of the two-stage design, and the significance of the EYE4ALL dataset for inclusive multimodal evaluation. We address the raised concerns and questions below.
>
> **W1. Dependency on GPT-4o mini and potential dataset bias**
>
> We agree that using GPT-4o mini for response refinement introduces a dependency on a proprietary model and may reflect stylistic bias in the generated text. This choice was a practical compromise driven by resource limitations. Collecting large-scale human annotations involving blind and low-vision (BLV) participants requires significant time and accessibility support, which was not feasible within our resource constraints. Instead, GPT-4o mini was used to generate refined candidate responses that could serve as an initial cornerstone for further research on accessibility-aware AI systems.
>
> Importantly, we release the training set publicly to promote transparency and allow future researchers to analyze, re-annotate, or de-bias the responses if needed. While GPT bias may exist, we emphasize that our work could serve as a crucial foundation for lowering the entry barrier to developing barrier-free AI systems, especially in communities where high-quality human-labeled data is scarce.
>
> Moreover, the final evaluation benchmark set of 1K manually verified samples was reviewed by 25 sighted human annotators following a clear rubric. Each sample was validated for sufficiency, safety, and spatial accuracy, ensuring that GPT-generated artifacts do not dominate the dataset’s final form. Moreover, by using GPT-4o to refine responses from various models rather than generating them from scratch, the process helps minimize the introduction of bias. We will make this motivation and the dataset curation rationale clearer in the revision to avoid misunderstanding.
>
> **W2. Use of Ridge Regression in Stage 2 and its potential limitations**
>
> We appreciate the reviewer’s thoughtful observation regarding the potential limitations of using a linear Ridge Regression model for multi-objective learning. While we acknowledge that Ridge Regression cannot explicitly model highly nonlinear interactions among objectives, our experiments suggest that this simplicity does not hinder performance in practice. Across datasets such as VisionREW, Align-anything, and EYE4ALLMulti, we found that the objectives exhibit relatively mild nonlinear dependencies, and that a linear model can capture them effectively while maintaining strong stability and interpretability.
>
> In particular, on the VisionREW benchmark, which contains 59 fine-grained evaluation dimensions and exhibits significant heterogeneity in human preference annotations, Ridge Regression still achieved strong results with an average accuracy of 87.2%. To further validate this, we compared Ridge Regression against nonlinear alternatives, including a small MLP and a Random Forest model, under identical training conditions. The results are summarized below:
>
> | **Model Type** | **Average Accuracy (%)** | **Inference Complexity** |
> | --- | --- | --- |
> | **Ridge Regression (ours)** | 87.2 | O(d), d: hidden dimension |
> | MLP | **88.6** | O(d*100+100^2) |
> | Random Forest | 88.2 | O(T*D), T(=100): # trees, D(=10), maximum tree depth |
>
> These results show that while nonlinear mappings offer only marginal accuracy improvements, they also introduce higher computational cost, particularly under the 59-dimensional VisionREW setting. Ridge Regression, in contrast, provides a lightweight and scalable solution that maintains high predictive performance without sacrificing interpretability or efficiency.
>
> The focus of Stage 2 was to design a modular, asynchronous, and easily extensible framework, not to maximize representational complexity. This structure allows new objectives to be trained independently using frozen embeddings, avoiding the instability often seen in end-to-end fine-tuning. We will clarify in the revised manuscript that the adoption of Ridge Regression represents a deliberate design trade-off that balances efficiency, interpretability, and scalability, even under complex, high-dimensional evaluation conditions.

---

> ### Author Response · Authors · 2025-11-18
>
> **Q1. On the correlation between Overall Score (Stage 1) and Sufficiency (Stage 2)**
>
> We appreciate the reviewer’s insightful question. The correlation illustrated in Figure 9 originates from human annotations in the EYE4ALL dataset rather than from the model’s learning process. It reflects how humans often consider sufficiency, defined as the completeness of navigational information (for BLV users), to be a primary factor influencing the perceived overall quality of a response in accessibility-oriented evaluations.
>
> Although leveraging the correlations observed in the human experiment may be able to guide better training dataset generation, we claim that the training dataset should contain a large-scale image-text pairs dataset scored with a diverse score range for each criterion to avoid overfitting for one particular dimension. As shown in Table 10, we carefully prompt GPT to generate the corresponding request, response, and scores for 7 evaluation criteria. Subsequently, the resulting training dataset consists of a wide range of scores for each criterion, serving as a good starting point for training the multi-objectives of MULTI-TAP to be evaluated on EYE4ALL. We are actively investigating how to construct higher-quality training data to further improve optimization, a time-intensive effort that we plan to pursue in future extensions of our work.
>
> We once again thank the reviewer for the constructive comments. The feedback helped us clarify the design motivation of our dataset, the rationale behind the lightweight regression module, and the broader implications of the human-annotated correlations observed in EYE4ALL. All clarifications and additional context will be incorporated into the revised version. We would truly appreciate it if you could reconsider the score if the points have been clarified. Thank you again for your time in providing helpful feedback.

---

> > ### Author Response · Authors · 2025-11-27
> > **Gentle Reminder**
> >
> > Dear Reviewer mKHf,
> >
> > As the review period is coming to an end, we wanted to reach out once more with our appreciation for your helpful feedback to improve our paper. We hope our responses have addressed the points you raised. If there is any further clarification needed for your final assessment, please feel free to let us know. Thank you.
> >
> > Kind regards,
> >
> > The Authors

---

### Author Response · Authors · 2025-11-28
**General Response**

We sincerely thank all reviewers for their valuable time and constructive feedback. With less than a week remaining in the review period, we would like to briefly follow up to provide this general response, clarifying our main contributions and summarizing how we have addressed common concerns.

Please check the updated PDF file, where we have highlighted the changes in **blue**. We also added Table 5 to better justify the choice of ridge regression for Stage 2.

> ## **Main Contributions**

 **MULTI-TAP: A plug-and-play robust LVLM-based predictor**:
Our predictor consists of two stages, where the first stage outputs an overall image-text alignment score, and the second stage returns fine-grained scores across human-interpretable dimensions. Due to the simplicity of our method, our MULTI-TAP is model-agnostic, easily applicable to different  LVLM architectures.

**High human-alignment and efficiency**:
MULTI-TAP achieves strong correlation with human judgments across diverse datasets for both the overall alignment score and multiple scores across various dimensions, with substantially reduced inference cost compared to conventional generative models, including GPT-4o, and a recent high-performing multi-objective reward model, VisionREWARD.

**Multi-objective scoring for real-world applicability**:
MULTI-TAP predicts dimension-specific scores (accuracy, safety, sufficiency, etc.) essential for **accessibility-critical contexts**. We test this on our novel, human-validated dataset, Eye4ALL, consisting of human and BLV user preferences annotated across seven evaluation dimensions.

**Transparency**:
Importantly, we publicly release all the datasets and detailed model training procedures, and checkpoints to ensure transparency and encourage future research on the underexplored area of investigating the preferences of non-sighted individuals.

> ## **Revisions and Clarifications**

**Multi-objective scoring stage value**:
 We emphasize that the multi-objective branch is not an auxiliary add-on but a central part of MULTI-TAP’s practical usability. While the scalar head provides an overall alignment score, real-world evaluation, especially in accessibility and safety-sensitive settings, requires interpretable, fine-grained scores across multiple human-relevant dimensions (accuracy, safety, sufficiency, etc.).

**Choice of ridge regression**:
Ridge regression was chosen to balance simplicity, interpretability, and computational efficiency. Our ablations show that nonlinear alternatives (small MLPs, Random Forests) provide only marginal accuracy gains at the cost of higher computation and instability. On VisionREW (59 dimensions), Ridge Regression achieves 87.2% accuracy, demonstrating that the Stage 1 embeddings already encode sufficiently rich multimodal structure for high-dimensional prediction.

**Dataset and performance comparisons**:
 We strictly adhered to train/test splits of Polaris, ImageReward, and other benchmarks. Observed performance gains generalize across heterogeneous, unseen datasets (FlickrExp, FOILR, PASCAL-50S, Sightation, EYE4ALL), indicating that improvements stem from the scalar reward-head framework rather than dataset overlap. Comparisons with IXCREW-S and other generative judges account for differences in evaluation paradigms; MULTI-TAP demonstrates robust ranking alignment under both pointwise and pairwise metrics while dramatically reducing inference cost.

**EYE4ALL dataset and GPT-4o mini refinement**:
 While GPT-4o mini was used to refine candidate responses for efficiency, the final evaluation benchmark consists of 1K manually verified samples reviewed by 25 human annotators for sufficiency, safety, and spatial accuracy. The public release of the training set ensures transparency and allows re-annotation or debiasing by future researchers. This dataset represents an important contribution toward inclusive, BLV-oriented multimodal evaluation.

**Stage 1 and stage 2 interaction**:
We clarify that the two-stage design is complementary: Stage 1 establishes generalizable, human-aligned embeddings, while Stage 2 translates these embeddings into fine-grained, interpretable scores. This modular structure prioritizes extensibility, stability, and efficiency. Although Stage 2 does not directly influence Stage 1 embeddings during training, it leverages their semantic richness to deliver a multi-objective evaluation that aligns with human judgments across multiple dimensions.

> ## **Conclusion**

MULTI-TAP introduces a modular, efficient, and interpretable framework for multi-objective evaluation of LVLM outputs. By combining a scalar reward head with lightweight regression for fine-grained scoring, it achieves strong human alignment, robust generalization across datasets, and practical applicability in accessibility- and safety-critical settings. We hope these clarifications address the reviewers’ concerns and would greatly appreciate reconsideration of the scores.

---

### Meta-Review · Area_Chair_D8xx · 2025-12-29

**Summary:**

In this paper, the authors propose MULTI-TAP, which learns a reward aligned with human judgments using MSE loss, and then adds a lightweight ridge regression layer to generate multi-objective scores. They also present a new dataset, called EYE4ALL, which captures human preferences across seven evaluation dimensions. Finally, they empirically show that MULTI-TAP achieves stronger correlation with human ratings than existing metrics, such as CLIP-S and IXCREW-S, performs competitively with G-VEval, and surpasses VisionREW-S on multi-objective evaluation tasks. Of course, I do not think observing similar performance with G-VEval is surprising, given the GPT-4o mini bias in the EYE4ALL dataset.

The reviewers appreciate the new dataset EYE4ALL and mostly agree that several claims are well supported by experiments. I believe that the extra results provided by the authors during the rebuttals (MLP and Random Forest) address the reviewers' concern on the limitations of ridge Regression to learn multi-objective scores. However, the paper does not do a good job in motivating the need for both single- and multi-objective scores, especially when they are highly correlated with each other. Moreover, the reviewers feel that the paper can benefit from ablation studies and other experiments that more clearly show the source of performance gain of the proposed method.

Although the average score is below acceptance, I see this as a borderline paper. This is why I strongly encourage the authors to revise their paper to address the reviewers' comments and prepare it for upcoming venues.

**Reviewer Concerns:**

I believe the authors' response addresses most of the reviewers' concerns, except the motivation and role of having both multi and single objective scores, and the interplay between them. This is an important concern and the authors should properly justify why both these scores are needed and their potential applications. Another concern that was remained partially unanswered is the influence of GPT-4o mini on the EYE4ALL dataset, which creates a bias toward GPT models in this dataset.

**Reviewer Scores:**

Reviewer hQRF did not participate in the discussion. The responses addressed the concerns of Reviewers yUqU and mKHf, but they decided to keep their score of 6 unchanged. Finally, the main concern of Reviewer sS7f remained unresolved, and thus, they decided to maintain their score of 2.

---

### Decision · Program_Chairs · 2026-01-26

Reject